# Outside-in engineering of cadherin endocytosis using a conformation strengthening antibody

Bin Xie [1,3], Shipeng Xu [2,3] & Sanjeevi Sivasankar [1,2] ✉

P-cadherin, a crucial cell-cell adhesion protein which is overexpressed in numerous malignant cancers, is a popular target for drug delivery antibodies. However, molecular guidelines for engineering antibodies that can be internalized upon binding to P-cadherin are unknown. Here, we use a combination of biophysical, biochemical, and cell biological methods to demonstrate that trapping the P-cadherin extracellular region in an X-dimer adhesive conformation triggers cadherin endocytosis via an outside-in signaling mechanism. We show that the anti-cancer drug delivery monoclonal antibody CQY684, traps P-cadherin in an X-dimer conformation and strengthens this adhesive structure. Formation of stable X-dimers results in the phosphorylation of p120-catenin, a suppressor of cadherin endocytosis. This triggers the dissociation of p120-catenin from the X-dimer cytoplasmic region, which increases P-cadherin turnover and targets the cadherin-antibody complex to the lysosome. Our results establish an outside-in signaling mechanism that provides fundamental insights into how cells regulate adhesion and that can be exploited by anti-cadherin antibodies for intracellular drug delivery.

P-cadherin (placental cadherin; Pcad), a member of the classical cadherin family of cell-cell adhesion proteins, is overexpressed in malignant cancers[1], including cancers that originate from the breast[2], pancreas[3] and lung[4]. Consequently, Pcad is recognized as a promising monoclonal antibody (Mab) target for anti-cancer drug delivery[5]. In therapeutic applications, drug conjugated Mabs specifically target the extracellular regions of adherent Pcads at cell-cell junctions[6]. Upon binding to Pcad ectodomains, the Mabs are internalized for payload release[7]. However, the molecular linkage between extracellular cadherin conformations and cadherin endocytosis is unknown. Consequently, molecular guidelines for engineering Mabs that bind to Pcad ectodomains and promote cadherin internalization have not been established.

Like all classical cadherins, Pcad ectodomains bind in two distinct adhesive conformations: X-dimers and strand-swap dimers (S-dimers)[8,9]. Due to their faster on-rate and weaker binding strength, X-dimers are thought to be an intermediate in the formation and dissociation of

S-dimers[9–11]. Notably, cells expressing mutant cadherins trapped in an X-dimer conformation, exhibit more dynamic cell junctions due to increased cadherin turnover[10]. However, the mechanism by which X-dimer formation promotes cadherin internalization is unknown.

A master regulator of cadherin internalization is p120-catenin (p120), a protein that binds to the cadherin cytoplasmic region[12]. By acting as an endocytosis inhibitor, p120 stabilizes cadherins on the cell surface. Conversely, the dissociation of p120 triggers cadherin endocytosis and promotes their internalization[13]. However, it is unclear if X-dimer formation triggers cytoplasmic p120 dissociation and promotes cadherin endocytosis. Studying this outside-in mechanism is challenging due to a lack of methods to trap cadherins in an X-dimer conformation without introducing mutations into the cadherin extracellular region.

One strategy to trap cadherins in distinct binding conformations is the use of Mabs. For instance, two E-cadherin (Ecad) specific Mabs, 19A11 and 66E8, have been shown to bind and stabilize S-dimers[14–16].

[1]Biophysics Graduate Group, University of California, Davis, CA, USA. [2]Department of Biomedical Engineering, University of California, Davis, CA, USA. [3]These authors contributed equally: Bin Xie, Shipeng Xu. ✉e-mail: ssivasankar@ucdavis.edu

The stabilization of Ecad S-dimers on the cell surface has been shown to strengthen cell adhesion and promote the dephosphorylation of p120[17,18], suggesting a possible outside-in relationship between Ecad extracellular binding conformations and intracellular signaling. Conversely, the Mab CQY684, part of the antibody-drug conjugate PCA062, was recently shown to recognize the ectodomains of Pcad. Binding of CQY684 resulted in the endocytosis of the Mab-Pcad complex, thereby facilitating drug delivery[19]. Despite the potential of CQY684 as a drug delivery platform and its usage in Phase-I clinical trials[20], the molecular mechanism by which CQY684 triggers Pcad endocytosis is unknown.

In this study, we combine biophysical (atomic force microscopy, and bead aggregation), computational (molecular dynamics and steered molecular dynamics simulations), biochemical (surface biotinylation, phos-tag gels, and co-immunoprecipitation), and cell biological (cell adhesion assays, confocal imaging, and fluorescence recovery after photobleaching) measurements to demonstrate that CQY684 traps Pcad in an X-dimer conformation and stabilizes this binding structure. We show that the stabilization of X-dimers triggers the phosphorylation of p120 which results in its dissociation from the Pcad cytoplasmic region. The dissociation of p120 targets the antibody-Pcad complex to the lysosome. Our results establish an outside-in signaling mechanism that provides fundamental insights into how cells regulate adhesion. Cadherin outside-in signaling can also be exploited by antibodies for intracellular drug delivery.

## Results

### CQY684 selectively strengthens Pcad X-dimers

Since a previous structural study has shown that CQY684 binding does not interfere with formation of either Pcad S-dimers or X-dimers[19], we used atomic force microscopy (AFM) to measure the effect of CQY684 on the stability of these distinct Pcad conformations. We used three constructs in our experiments: human Pcad W2A mutant (trapped in an X-dimer conformation)[9], human Pcad K14E mutant (trapped in an S-dimer conformation)[9], and wild-type human Pcad (WT). We immobilized the complete extracellular region of each Pcad construct (EC1–5) on an AFM cantilever and glass substrate functionalized with polyethylene glycol (PEG) tethers, and measured Pcad–Pcad interactions with or without 40 nM CQY684 in the buffer ('+CQY' or '−CQY', Fig. 1a, upper panel).

During a typical AFM measurement, the cantilever and substrate, both functionalized with Pcad, were brought into contact to allow cadherin interactions. The tip was then retracted at a constant velocity (1 μm/s) to measure the force required to rupture the Pcad–Pcad bond. Unbinding events, characterized by nonlinear stretching of PEG tethers, were analyzed using a worm-like chain (WLC) model with least-squares fitting (Fig. 1a, lower panel). We confirmed that CQY684 recognizes all the mutants using Western blots (Supplementary Fig. 1). Unbinding force histograms were fitted to Gaussian distributions, with the optimal number of distributions predicted using the Bayesian Information Criterion (BIC, Supplementary Fig. 2).

For Pcad W2A mutants which can only form X-dimers, a single Gaussian distribution with a peak force of $24.5 \pm 9.9$ pN was measured in the −CQY condition. In the +CQY condition, W2A binding was strengthened and two Gaussian distributions with peak forces of $35.1 \pm 9.7$ pN and $55.8 \pm 7.5$ pN were measured (Fig. 1b, Supplementary Fig. 2a), indicating that CQY684 enhances Pcad X-dimer binding strength. To confirm that the measured bimodal force distribution was not an artifact of pulling velocity, we performed the same experiments with a higher pulling velocity (5 μm/s). At the higher pulling velocity, we again measured a single Gaussian force distribution without CQY684 and a bimodal force distribution in the +CQY condition (Supplementary Fig. 3). Atomistic computer simulations (Figs. 2, 3) demonstrated that CQY684 binding not only directly strengthened adhesion by

stabilizing the X-dimer interface, but also stochastically strengthened adhesion by inducing formation of a salt bridge between the two partner cadherins. The stochastic nature of the salt-bridge formation resulted in the bimodal distribution of unbinding forces observed with X-dimers.

For Pcad K14E mutants, which can only form S-dimers, a single gaussian unbinding force distribution was measured in both −CQY and +CQY conditions with similar peak forces ($33.1 \pm 13.4$ pN and $31.0 \pm 10.2$ pN, respectively) (Fig. 1c, Supplementary Fig. 2b), suggesting that CQY684 does not affect Pcad S-dimer binding strength. To eliminate the possibility that the K14E mutant has a low affinity for CQY684 and requires a higher Mab concentration to get strengthened, we also performed the +CQY AFM experiment at a higher, 500 nM CQY684 concentration. Even at this higher Mab concentration, the measured unbinding force histogram was best fit to a single Gaussian and no CQY684 mediated strengthening was observed (Supplementary Fig. 4).

Although WT cadherin ectodomains can adopt either X-dimer or S-dimer conformation in solution, X-dimers are believed to serve as a transient intermediate and cadherins are known to ultimately form stable S-dimers. Indeed, for Pcad WT in the -CQY condition, we observed a single Gaussian force distribution with a peak force of $33.1 \pm 12.4$ pN (Fig. 1d, Supplementary Fig. 2c upper panel), similar to Pcad K14E but higher than Pcad W2A, suggesting that Pcad WT primarily forms S-dimers. However, in the presence of CQY684, two Gaussian force distributions were measured with peaks at $34.2 \pm 13.2$ pN and $49.7 \pm 20.2$ pN (Fig. 1d, Supplementary Fig. 2c lower panel), similar to the force distribution of Pcad W2A in the +CQY condition. Since CQY684 only strengthens X-dimers and does not affect S-dimers, we concluded that the observed strengthening in the WT + CQY condition (Fig. 1d) was due to CQY684 trapping the WT Pcad in an X-dimer structure.

In addition to AFM experiments, we performed bead aggregation assays on the three Pcad constructs, with and without CQY684. While AFM experiments provide precise measurements of individual interactions, bead aggregation assays assess how CQY684 affects Pcad ectodomain binding in a more collective manner. In both the W2A mutants and WT, the addition of CQY684 increased the size of the aggregates (Fig. 1e, f), suggesting that CQY684 enhances Pcad- adhesion when the X-dimer state is accessible. The size of bead aggregates in Pcad WT + CQY was significantly larger than Pcad W2A + CQY due to the formation of both WT S-dimers and stabilized WT X-dimers. Consistent with previous studies[8,9], Pcad K14E mutants did not form bead aggregates, likely due to the low on-rate of K14E mutants, and the addition of CQY684 did not increase K14E aggregation (Supplementary Fig. 5).

In summary, our cell-free experiments revealed that CQY684 selectively strengthens Pcad X-dimers without affecting S-dimers. The observed strengthening effect with WT cadherin indicates that CQY684 traps Pcad in an X-dimer structure.

### Molecular mechanism of CQY684 mediated stabilization of X-dimers

The crystal structure of CQY684 bound to the Pcad EC1-2 domain demonstrates that Pcad bound to the antibody adopts an X-dimer conformation[19] (Fig. 2a, upper panel). This is not surprising since the Pcads in the crystal structure possess an N-terminal extension and preferentially form X-dimers[9]. CQY684 directly binds to three loops of the Pcad EC1 domain (Fig. 2b), referred to as 'loop1' (residues 15-17), 'loop2' (residues 43-51), and 'loop3' (residues 63-65). The binding interface of CQY684 is situated between the Pcad X-dimer interface, which is composed of residues 5–14, 20–23, 99–106, 138–145, and 196–204. Our previous studies have shown that antibodies binding near cadherin binding interfaces can stabilize these interfaces, thereby strengthening adhesion[14,15]. To test if CQY684 stabilizes the

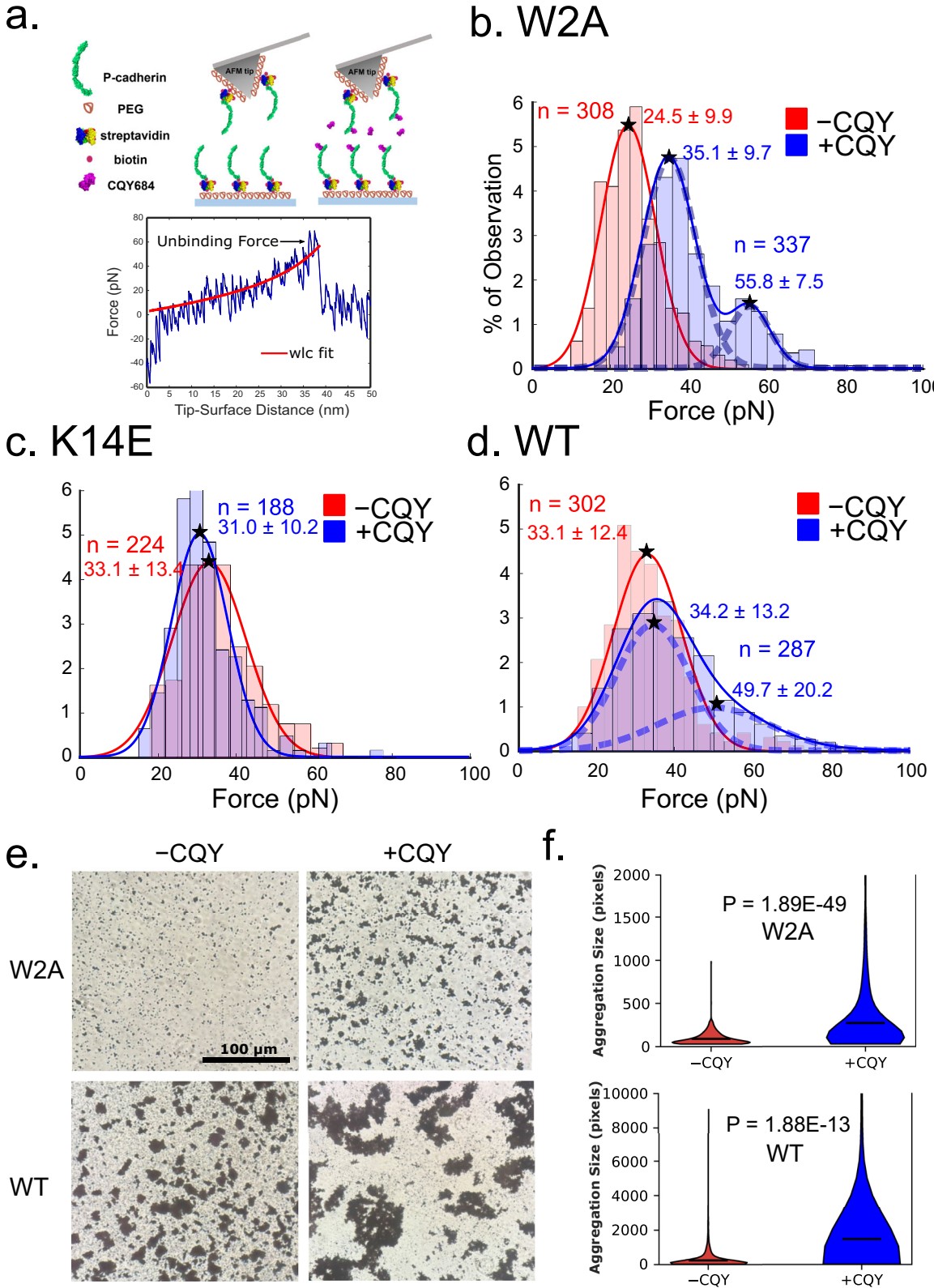

X-dimer binding interface, we performed molecular dynamics (MD) simulations on Pcad X-dimers under two conditions: '+CQY' with two CQY684 Fabs bound to each Pcad in an X-dimer conformation (PDB 6ZTR), and '-CQY' obtained from the same crystal structure by removing the CQY684 Fabs (Fig. 2a).

Our MD simulations, set up as previously described[14,15], included five independent repeats for each condition, with each repeat running

for 60 ns until the system reached equilibrium (Supplementary Fig. 6). To evaluate whether CQY684 stabilizes the Pcad X-dimer, we measured the root mean square fluctuations (RMSF) of the α-carbon residues during the last 30 ns of all MD simulations (Fig. 2c). The average RMSF for Pcad with and without the antibody showed that CQY684 binding not only reduced the RMSF in its binding regions on Pcad (Fig. 2c, red regions) but also stabilized the entire Pcad EC1 and EC2

**Fig. 1 | CQY684 Fab strengthens Pcad X-dimers. a** Upper panel: scheme for AFM experiments carried out in the absence of CQY684 Fab (−CQY), and in the presence of CQY684 Fab (+CQY). Full ectodomains of Pcad were immobilized on AFM tips and substrates functionalized with PEG tethers. Bottom panel: example force curve. Stretching of the PEG tether was fit to a WLC model (red line). **b**–**d** AFM Experiments were performed with W2A, K14E and WT Pcad in the absence (−CQY, red curves and histograms) and presence (+CQY, blue curves and histograms) of 40 nM CQY684 Fab. Histograms of the unbinding forces were generated by binning the data using the Freedman–Diaconis rule. The optimal number of Gaussian distributions for each fit was determined using BIC. N corresponds to the number of measured single molecule unbinding events. **b** In -CQY condition, W2A Pcad unbinding forces were best described by a single Gaussian ($24.5 \pm 9.9$ pN). Addition of CQY684 strengthened X-dimers; unbinding forces increased and the distribution was best described by two Gaussians at higher peak forces ($35.1 \pm 9.7$ pN and

$55.8 \pm 7.5$ pN respectively). **c** Unbinding force distributions measured with K14E Pcad were similar in both the -CQY and +CQY condition demonstrating that CQY684 did not strengthen S-dimers. **d** Unbinding force distributions measured with WT Pcad in the absence of CQY684 was best fit by a single Gaussian ($33.1 \pm 12.4$ pN corresponding to S-dimer conformation). Addition of CQY684 selectively strengthened the X-dimer and resulted in a bimodal force distribution ($34.2 \pm 13.2$ pN and $49.7 \pm 20.2$ pN respectively). **e** Bead aggregation assays showed that beads functionalized with either W2A or WT Pcad formed larger aggregates in the presence of CQY684 Fab. Scale bar: 100 μm. **f** Violin plots of bead aggregate sizes. Mean size of the aggregate is show as a black line on each violin. Two independent biological repeats were performed for bead aggregation experiments. $N = 1538$ bead aggregates (W2A -CQY), 2113 aggregates (W2A + CQY), 2069 aggregates (WT -CQY), and 405 aggregates (WT + CQY). Two-sided student t-test was performed on the violin plot.

domains, including the X-dimer binding interface (Fig. 2c, blue regions).

Additionally, we characterized the electrostatic interactions (i.e. hydrogen bonds or salt bridges) between Pcads in all simulations (Supplementary Fig. 7). Although CQY684 binding did not induce a gross conformational change in the X-dimer, we observed that persistent interactions between Pcads (i.e. interactions that existed for at least 40% of simulation time), changed upon CQY684 binding. Significantly, a previously non-persistent salt bridge which was present in the X-dimer binding interface (105Lys:199Asp), became persistent upon the binding of CQY684 (Fig. 2d). To monitor the formation of this salt bridge, we calculated the distance between charged atoms. Using the criterion that salt bridges form when the median distance between charged atoms is below 4Å, our analysis showed that in the presence of CQY684 the 105Lys:199Asp salt bridge formed in three sets of simulations (sets 3–5, Fig. 2e).

Although the conversion between cadherin X-dimers and S-dimers is not fully understood, we have previously shown that external pulling forces can convert X-dimers into S-dimers[21]. To test if the stabilized X-dimer induced by CQY684 binding inhibits this conversion, we performed Steered Molecular Dynamics (SMD) simulations on all the equilibrium structures from the MD simulations. In our SMD setup, we fixed the position of one Pcad and applied a constant force to pull the C-terminus of the other Pcad until the structures fully dissociated. To monitor the conversion of an X-dimer to an S-dimer structure, we calculated the root mean square deviation (RMSD) of the SMD structures relative to the Pcad S-dimer crystal structure (PDB code: 4ZMN). Independent of CQY684 binding, all X-dimers showed a conversion towards S-dimers before fully dissociating, upon application of pulling force (Fig. 3a, b; Supplementary Movie 1). However, in the absence of CQY684, the average conversion time was $1012 \pm 212$ ps, while CQY684 binding dramatically slowed this conversion nearly five-fold, with an average conversion time of $5307 \pm 2355$ ps (Fig. 3b). Given that previous studies suggest cadherin X-dimers are a necessary intermediate for S-dimer formation[8,9,22,23], and our results show CQY684 inhibits the Pcad X-dimer transition to S-dimer, it is likely that CQY684 traps the interacting Pcad in an X-dimer conformation. This could explain why AFM measurements show that Pcad WT + CQY has a force distribution similar to W2A + CQY but different from K14E + CQY (Fig. 1b–d).

To test whether the stabilized X-dimer induced by CQY684 binding could lead to stronger interactions as observed in the AFM experiments, we measured the X-dimer dissociation time during each SMD simulation. Specifically, we estimated the interfacial binding area between the two Pcads during SMD by calculating the change in solvent accessible surface area (ΔSASA), where a decrease in ΔSASA to zero corresponds to the rupture of the interacting dimer (Fig. 3c). In the absence of CQY684, the average dissociation time was $1207 \pm 204$ ps, while in the presence of CQY684, the average dissociation time increased to $5762 \pm 2482$ ps. Notably, when the novel

salt bridge (105Lys:199Asp) stably formed between Pcads in sets 3–5, the average dissociation time of sets 3–5 (~7300 ps, Fig. 3c, dark blue) was almost two-fold longer than that of sets 1 and 2 (~3500 ps, Fig. 3c, light blue). This could explain why the AFM unbinding force histograms in the W2A + CQY and WT + CQY conditions are bimodal (Fig. 1b, d) with the lower force distribution corresponding to instances where the salt bridge is not formed and the higher force distribution corresponding to cases when the salt bridge is formed.

To further investigate the mechanistic basis by which CQY684 strengthens X-dimers, we mapped changes in electrostatic interactions (hydrogen bonds and salt bridges) in the presence of CQY684 but without the persistent salt bridge (+CQY -salt bridge), in the presence of CQY684 with the persistent salt bridge (+CQY +salt bridge), and in the absence of CQY684 (-CQY) (Fig. 3d). A notable difference between the "+CQY -salt bridge" and "+CQY +salt bridge" conditions during the SMD simulation was the duration of the 105Lys:199Asp salt bridge. In the "+CQY -salt bridge" condition, this salt bridge forms transiently at the start of the simulation, whereas in the "+CQY +salt bridge" condition, it persisted for ~6 ns, significantly strengthening the hydrogen bonds at the X-dimer interface. This analysis suggests that the formation of the 105Lys:199Asp salt bridge is responsible for the second, higher force Gaussian in the bimodal X-dimer unbinding histograms measured with the AFM.

In summary, our computational studies revealed that CQY684 binding stabilizes the Pcad X-dimer conformation, preventing its conversion to an S-dimer structure. Furthermore, the stabilized X-dimer results in much stronger interactions, consistent with our AFM and bead aggregation results.

## CQY684 promotes Pcad endocytosis and disrupts Pcad-mediated cell adhesion

Cell-cell adhesion plays a crucial role in cancer progression; down-regulation of cell adhesion occurs in cancer metastasis[24] while upregulation of adhesion prevents metastasis[25]. However, the impact of CQY684 on Pcad-mediated cell-cell adhesion has not been previously characterized. Given that CQY684 strengthens Pcad X-dimers at the single molecule level and enhances bead aggregation with Pcad WT and W2A mutants, we anticipated that CQY684 would also enhance Pcad-mediated cell-cell adhesion. To test this, we performed cell adhesion assays on cells expressing either WT or W2A mutant Pcad. Specifically, we rescued Ecad and Pcad knockout (KO) A431 cells[26] with either full length WT-Pcad or W2A-Pcad with an mCherry tag on the Pcad C-terminus.

To test the effect of CQY684 on Pcad-mediated cell-cell adhesion, we performed a dispase assay to evaluate cell adhesion strength under mechanical stress. In this assay, the cells expressing either WT-Pcad or W2A mutants were grown to 95% confluence, with 200 nM CQY684 Fab added during overnight growth for the +CQY conditions. Using immunofluorescence imaging, we confirmed that CQY684 Fabs were localized to the cell surface during the time course of the experiments

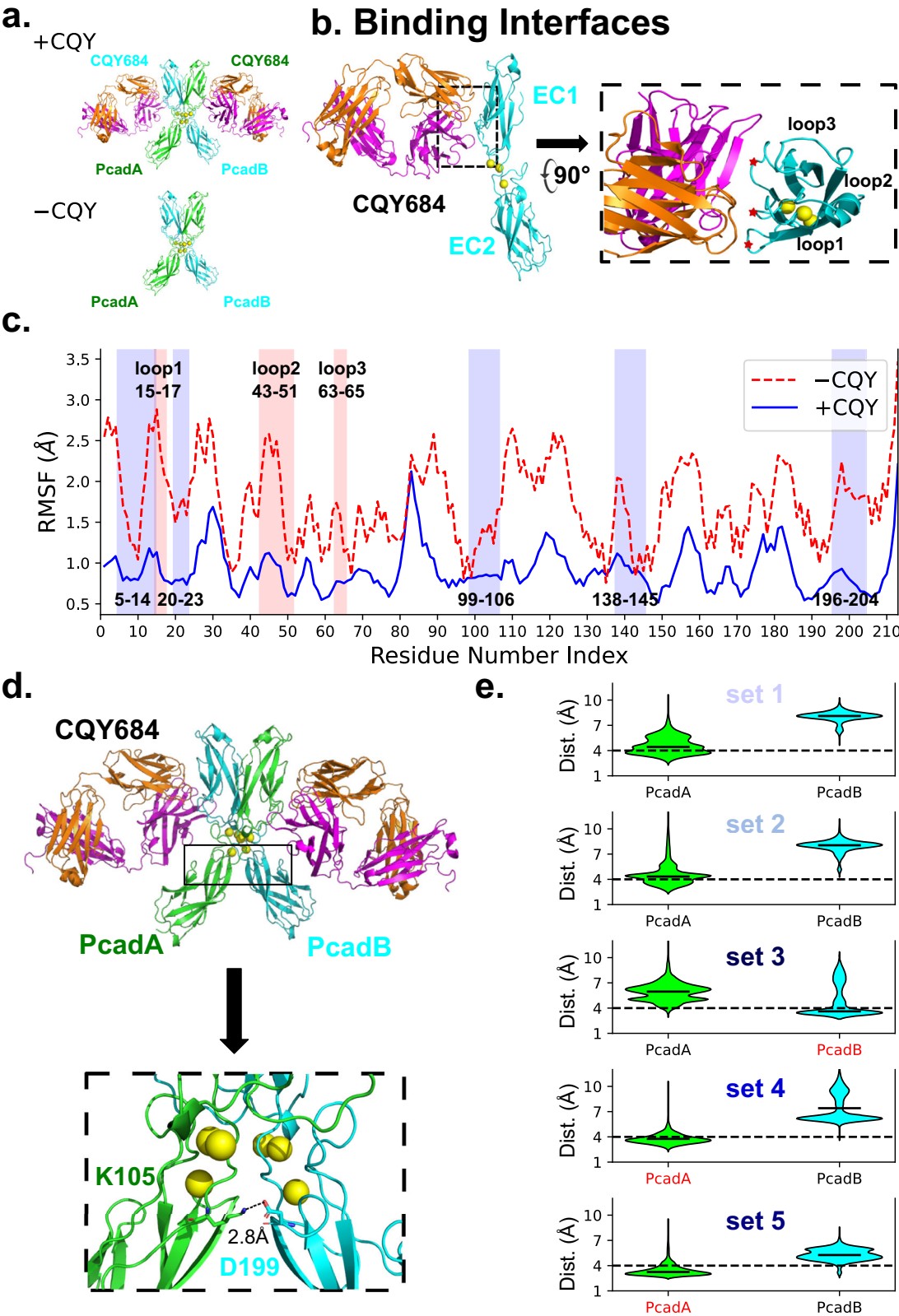

**a.** +CQY / −CQY panels: CQY684, PcadA, PcadB

**b. Binding Interfaces** — CQY684, EC1, EC2, loop1, loop2, loop3, 90°

**c.** RMSF (Å) vs Residue Number Index; −CQY (red dashed), +CQY (blue solid); loop1 15-17, loop2 43-51, loop3 63-65; 5-14, 20-23, 99-106, 138-145, 196-204

**d.** CQY684, PcadA, PcadB; K105, D199, 2.8Å

**e.** Dist. (Å) violin plots: set 1, set 2, set 3, set 4, set 5; PcadA, PcadB

(Supplementary Fig. 8). The cell monolayers were then detached from the dish using dispase enzyme and subjected to mechanical stress by vigorously rotating them on an orbital rotator for 2 h. This process applied shear forces to the cell layers, allowing us to assess how adherent the cells remained under stress, both in the absence and presence of CQY684. Surprisingly, the monolayers treated with CQY684 showed significantly more fragmentation compared to those

without CQY684 treatment (Fig. 4a). Contrary to our expectations based on the bead aggregation results, the addition of CQY684 weakened cell-cell adhesion rather than strengthening it.

In addition to the dispase assay, we also evaluated cell-cell adhesion using a cell aggregation assay. Consistent with the dispase assay results, A431 cells expressing WT-Pcad formed smaller aggregates in the presence of CQY684 (Fig. 4b), indicating that CQY684 disrupts

**Fig. 2 | CQY684 stabilizes Pcad X-dimer binding interface. a** Simulation setup for the +CQY condition (with two CQY684 Fabs bound to an X-dimer) and the −CQY Pcad X-dimer. **b** CQY684 Fab binds to the Pcad EC1 domain by interacting with three loops. The position of the three loops are indicated with star symbols. **c** Comparison of average RMSF values for Pcad residues 1-213 in the −CQY condition (red dashed line) and in the +CQY condition (blue solid line). The CQY684 binding interface is highlighted with a red background while the X-dimer interface is highlighted using a blue background. The lower RMSF values in the +CQY condition shows that the binding of CQY684 stabilizes the three loops of Pcad and also stabilizes the X-dimer binding interface. **d** CQY684 binding induces formation of a novel salt bridge between K105 and D199, within the X-dimer binding interface. The representative structure shown is the final structure of +CQY set 5. **e** Violin plots of distances between the charged atoms in the 105LYS:199ASP salt bridge measured during the last 40 ns of each +CQY MD simulation. The median distance is shown as a blue line on each violin. If the median distance is below the 4Å (black dashed line), then the salt bridge is considered to be stably formed during MD simulations. Salt bridge donor on PcadA is shown on the left while salt bridge donor on PcadB is shown on the right. Set 3 has one stable salt bridge with the donor from PcadB, while sets 4 and 5 each have one stable salt bridge with the donor from PcadA, all highlighted in red. No salt bridge formation was observed for sets 1 and 2.

Pcad-mediated cell-cell adhesion. Notably, A431 cells expressing Pcad W2A mutants showed extremely low cell adhesion in both assays: in the dispase assay, cells could not be lifted as a single monolayer, and in the cell aggregation assay, no aggregates were observed with or without CQY684 (Fig. 4b). The loss of adhesion in Pcad W2A mutants is likely due to the highly dynamic junctions formed by these mutants[10].

Since a previous study shows that X-dimer formation increases cadherin turnover[10], we hypothesized that CQY684 traps WT-Pcad in an X-dimer conformation, which consequently enhances Pcad turnover. To test this hypothesis, we performed immunofluorescence imaging using a confocal microscope. A431 cells expressing WT-Pcad were grown to 70% confluency, with 200 nM CQY684 Fab added during cell growth for the +CQY condition. Since a signature of increased cadherin turnover is the internalization of cadherins and their subsequent colocalization with lysosomes, we stained for both Pcad and lysosomal-associated membrane protein 1 (LAMP1), a well-established lysosomal marker[27]. Cells treated with CQY684 Fab showed increased Pcad internalization and high colocalization with LAMP1 (Fig. 4c), suggesting that CQY684 induces Pcad turnover, and lowers Pcad surface expression levels, thereby disrupting Pcad-mediated cell-cell adhesion. Importantly, the integrated LAMP1 intensity within individual cells was the same for both the +CQY and −CQY conditions (Supplementary Fig. 9).

To investigate changes in Pcad surface expression, we performed surface biotinylation experiments with WT -CQY, WT + CQY, and W2A cells. Detergent compatible (DC) protein assays were used to normalize the protein amounts for the western blot analysis. As expected, the surface level of Pcad in WT cells decreased upon CQY684 treatment (Fig. 4d, middle panel), likely due to increased internalization of Pcad and subsequent lysosomal proteolysis. While the W2A cells exhibited higher levels of surface Pcad compared to WT cells, likely due to higher transfection efficiency (Fig. 4d, middle panel), cell-cell adhesion between W2A cells was severely impaired. This suggests that factors beyond surface Pcad levels are responsible for adhesion defects in both WT + CQY and W2A conditions. Notably, the fraction of total Pcad expressed on the cell surface was similar (~35% of total Pcad) across WT -CQY, WT + CQY, and W2A conditions (Fig. 4d, right panel). Consistent with previous results[10], this suggested that neither the addition of CQY684 nor the W2A mutation affected Pcad localization to the cell membrane.

**p120-catenin phosphorylates and dissociates from the X-dimer cytoplasmic region, increasing junction dynamics**
Previous studies have shown that p120 phosphorylation and dissociation from the cadherin cytoplasmic domain signals endocytosis and increases cadherin turnover[13,28,29]. We therefore hypothesized that CQY684 binding would result in the phosphorylation and dissociation of p120 from the Pcad cytoplasmic domain. In addition, given our previous findings that CQY684 traps Pcad in an X-dimer conformation, we hypothesized that CQY684-independent X-dimer formation would also trigger p120 phosphorylation and dissociation from the cadherin cytoplasmic tail. To test if X-dimer formation, either via CQY684 or the W2A mutation, changes the phosphorylation state of p120, we performed phos-tag gel experiments in the WT − CQY, WT + CQY, and W2A conditions. We observed two well-separated bands in each lane of the Phos-tag gel: the top band corresponds to phosphorylated p120, and the bottom band represents unphosphorylated p120 (Fig. 5a, left panel). Compared to WT -CQY, we measured an ~15% increase in phosphorylated p120 in both WT + CQY and W2A conditions (Fig. 5a, right panel). These results suggest that X-dimers, either formed via CQY684 or the W2A mutation, trigger phosphorylation of p120.

To test if X-dimer formation results in p120 dissociation, we conducted co-immunoprecipitation (co-IP) experiments to measure the Pcad-p120 association in WT − CQY, WT + CQY, and W2A conditions (Fig. 5b). In the co-IP experiments, we used an anti-mCherry antibody to pull down Pcad as the bait and then measured the amount of p120 that co-precipitated with Pcad, normalizing these amounts to the WT − CQY condition. The WT + CQY showed ~60% of p120 associated with Pcad compared to the WT -CQY condition, a 40% decrease in p120 binding (Fig. 5b, right panel). Similarly, only ~30% of p120 were associated with the W2A mutant compared to the WT -CQY condition, confirming our hypothesis that X-dimer formation triggers p120 dissociation from the cadherin cytoplasmic tail (Fig. 5b, right panel). The measured change in p120 association/dissociation did not arise from differential Pcad expression, because the surface biotinylation assays show that similar fraction of Pcad localize on the cell surface in all the conditions (Fig. 4d, right panel).

To understand the effect of p120 dissociation on the dynamics at cell-cell junctions, we employed fluorescence recovery after photobleaching (FRAP), a technique that measures protein mobility in live cells. In the FRAP experiments, we used a laser to bleach a small region of mature adherens junction, resulting in the loss of mCherry fluorescence in that area. We then measured the time-dependent recovery in fluorescence signal which occurs due to the exchange of Pcad molecules (Fig. 5c). Since previous studies demonstrate that fluorescence recovery in mature adherens junctions primarily arise due to endocytosis and occur on the minute time scale[30,31], we monitored fluorescence recovery for ~3 min. Fluorescence recovery was fitted to an exponential equation, and the maximum fraction of fluorescence recovery (Fig. 5d, 'mobile fraction') and the half recovery time (Fig. 5d, '$t_{1/2}$') was obtained from the fitted curve (Fig. 5d, top panel). The maximum fraction of fluorescence recovery indicates the proportion of mobile proteins in that region with higher recovery signifying increased dynamics at the cell-cell junction while $t_{1/2}$ serves as a proxy for protein turnover rate, with shorter recovery time signifying more rapid turnover. For the A431 cells expressing WT Pcad, the mobile fraction was 38.7 ± 10.2% (Fig. 5d, bottom left panel, '−CQY'), whereas cells treated with CQY684 Fab exhibited a higher mobile fraction of 52.5 ± 11.4% (Fig. 5d, bottom left panel, '+CQY'). Consistent with previous results[10], cells expressing W2A Pcad also had a high mobile fraction (67.7 ± 12.4%; Fig. 5d, bottom left panel, 'W2A'). Similarly, both the W2A mutant and the WT + CQY had a shorter $t_{1/2}$ (35.9 ± 15.2 s and 41.9 ± 13.3 s respectively) compared with WT Pcad -CQY (56.0 ± 18.1 s; Fig. 5d, bottom right panel). Based on these FRAP results, we concluded that CQY684 increases the dynamics of cell-cell junctions by trapping Pcad in an X-dimer conformation, similar to the W2A mutant. Taken together, these results indicate that X-dimer formation triggers phosphorylation of p120 and its dissociation from the Pcad

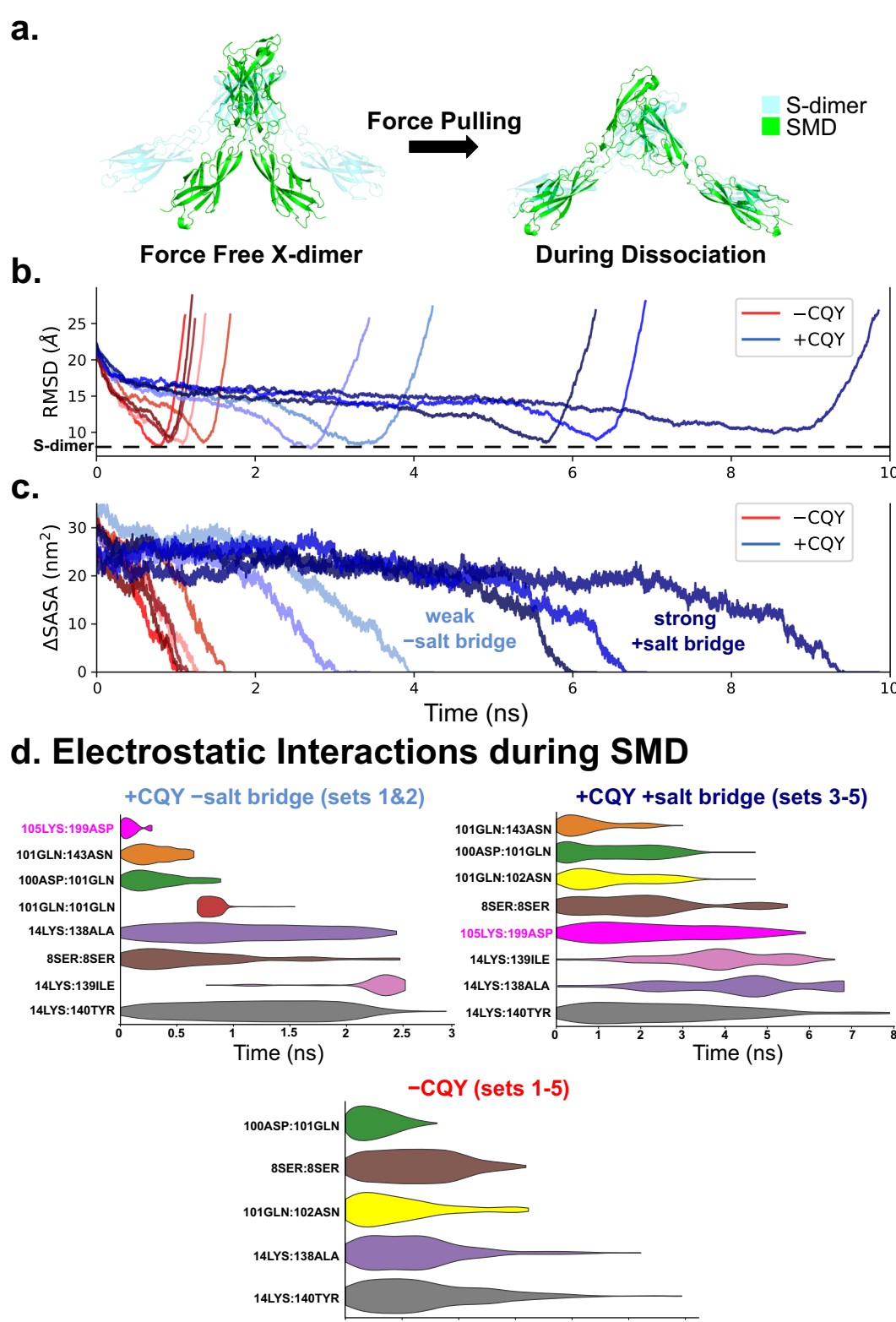

**d. Electrostatic Interactions during SMD**

cytoplasmic region. This facilitates cadherin endocytosis and weakens Pcad-mediated cell-cell adhesion (Fig. 5e).

## Discussion

Here, we describe an outside-in signaling mechanism that transduces cadherin ectodomain conformation into cytoplasmic signaling events. Specifically, we show that the Mab CQY684 traps and stabilizes the Pcad X-dimer conformation. This results in the phosphorylation of p120-catenin, and its dissociation from the cadherin cytoplasmic region, which triggers Pcad endocytosis. We anticipate that this overlooked outside-in signaling mechanism can be exploited for intracellular drug delivery. Rationally designing antibodies that are engineered to trap and stabilize X-dimers would target the antibody-cadherin complex to the lysosome for intracellular drug release.

**Fig. 3 | CQY684 inhibits Pcad transition from X-dimer to S-dimer and strengthens adhesion. a** While dissociating under a constant pulling force, Pcad X-dimers convert to S-dimers. Selected structures during SMD are showed in green. The S-dimer reference crystal structure is showed in cyan. **b** RMSD was calculated between every SMD structure and the reference S-dimer crystal structure. The RMSD values first drop (indicating that the X-dimer converts to an S-dimer like structure) and then increase (as the dimer dissociates). The addition of CQY684 (different shades of blue) dramatically slows down this structural conversion compared to the −CQY condition (different shades of red). **c** Pcad unbinding during the constant force SMD simulation was calculated from changes in interfacial area computed from the ΔSASA, in the absence of CQY684 (shades of red), and in the presence of CQY684 (shades of blue). The addition of CQY684 stabilizes the Pcad X-dimers and slows down their dissociation. When the salt bridge forms during the MD simulation (in sets 3−5, highlighted in darker blue), the Pcad interactions are significantly stronger compared to conditions (sets 1 and 2, highlighted in lighter blue), where the salt bridge does not form. **d** Violin plots showing the electrostatic interactions (salt bridges and hydrogen bonds) during the SMD simulations in the presence of CQY684 without/with the formation of a persistent 105Lys:199Asp salt bridge ("+CQY -salt bridge" / " + CQY +salt bridge"), and in the absence of CQY684 ("-CQY"). For clarity, only electrostatic interactions lasting >0.2 ns are shown. A notable difference between "+CQY -salt bridge" and "+CQY +salt bridge" was the duration of the 105Lys:199Asp interaction (highlighted in magenta), indicating that the persistent 105Lys:199Asp salt bridge substantially strengthens X-dimer interactions.

We have previously described how Ecad-activating monoclonal antibodies 19A11 and 66E8, which bind to the EC1 or EC2 domains, strengthen Ecad S-dimers by forming interactions with the K14 residue, thereby stabilizing the S-dimer binding interface. The K14E point mutation abolishes the strengthening effect of 19A11 and 66E8, emphasizing the critical role of this residue in S-dimer stabilization[14,15]. Additionally, since the K14 residue is involved in cadherin X-dimer formation, 19A11 and 66E8 trap Ecad in the S-dimer state. By preventing the transition from S-dimer to X-dimer, 19A11 and 66E8 may stabilize cadherin on the cell surface and inhibit turnover, thereby strengthening Ecad-mediated cell adhesion. In contrast, CQY684 binds to the Pcad EC1 domain but does not inhibit the formation of either X-dimers or S-dimers. Using cell-free experiments and computational simulations, we have demonstrated that CQY684 stabilizes the Pcad X-dimer interface, selectively strengthening the X-dimer without affecting the S-dimer, thus slowing the transition from X-dimer to S-dimer conformation. This selective strengthening of the X-dimer may be due to the lack of interactions with the K14 residue. The K14 residue could therefore serve as an important epitope for the further design of mAbs that regulate cadherin interactions.

PCA062, a drug-antibody conjugate that uses CQY684 linked to an anti-cancer drug, was previously found to be targeted to lysosomes for therapeutic drug delivery[19]. Our results demonstrate that CQY684 traps Pcad in an X-dimer conformation and subsequently promotes Pcad internalization. PCA062 had entered 'first-in-human' phase I clinical trials for treating Pcad-positive tumors. However, this trial was discontinued due to lack of efficacy and reported cancer progression across all participants[20]. One possible explanation for the failure of PCA062 is the overlooked effect of CQY684 disrupting cell adhesion. Loss of cell adhesion occurs during the early stages of cancer metastasis, and disrupting cell adhesion promotes cancer progression[24,32]. To avoid unexpected changes in cell adhesion, future experiments using CQY684 as a drug delivery platform should target cancer cell lines where Pcad is not the predominant cadherin.

In addition to addressing the overlooked effects of CQY684 on cell adhesion, our study resolves several previously published structural inconsistencies regarding the binding of CQY684 to Pcad[19]. It was originally suggested that when bound to CQY684, Pcad dimerizes in a unique asymmetric *trans* binding conformation that inhibits S-dimer formation[19]. However, the crystal structure (PDB 6ZTR) unambiguously demonstrates that CQY684 binds to the Pcad X-dimer conformation. This is in fact an expected result since the Pcad used in the crystal structure possess an N-terminal extension that is known to preferentially form X-dimers[9]. Consequently, the asymmetric binding conformation that was originally proposed[19] is likely a misinterpretation of the crystal structure. Furthermore, it was claimed that CQY684 disrupts Pcad *cis*-dimer formation due to steric collisions between the antibody binding interface and the *cis*-dimer interface[19]. However, an alignment of CQY684 on previously resolved Pcad *cis* dimer reveals no such steric interference suggesting that CQY684 does not disrupt *cis*-dimer formation (Supplementary Fig. 10).

In addition to its therapeutic potential, CQY684 can also serve as a powerful tool for studying the outside-in regulation of cadherins without introducing mutations in the cadherin ectodomain. We have shown that CQY684 traps Pcad in the X-dimer conformation, leading to the phosphorylation and dissociation of p120 from the cytoplasmic region, thereby promoting Pcad endocytosis and colocalization with lysosomal markers. However, while the Pcad W2A mutant also shows p120 phosphorylation and decreased p120 association, it does not colocalize with the lysosome (Supplementary Fig. 11a). This may be due to low proteolytic processing of the cytoplasmic region of the W2A mutant (Supplementary Fig. 11b), which is obligatory for cadherin endo-lysosomal trafficking[33]. Therefore, the W2A mutation not only traps cadherin in the X-dimer conformation but may also silence other signaling pathways, confounding studies on the effects of cadherin X-dimers in cells.

Previous studies suggest that X-dimers promote cadherin turnover on the cell surface by binding more weakly than S-dimers[10]. However, this explanation overlooks the outside-in signaling mechanism revealed by our results. Our data suggests that cadherins regulate cell adhesion, not merely by strengthening ectodomain binding, but also by using different binding conformations to signal association/dissociation of cytoplasmic proteins. Our results indicate that the signaling function of these adhesive conformations could be more crucial than a naïve interpretation based on differential binding strengths in solution.

Our study indicates that the strength of cell adhesion cannot be solely estimated by the amount of cadherins on the cell surface; associated adapter proteins like p120 must also be considered. To estimate the amount of cell surface Pcad associated with p120, referred to as "effective Pcad," we combined results from surface biotinylation and co-IP assays, assuming p120 only associates with surface-localized Pcad. We found that effective Pcad levels in both WT + CQY and W2A conditions were half of those in WT -CQY conditions (Supplementary Fig. 12). This reduction in effective Pcad likely explains the disruptions in cell adhesion caused by CQY684 or the W2A mutations, rather than just changes in surface Pcad levels. Notably, WT + CQY has a similar amount of effective Pcad as W2A but stronger cell adhesion, possibly because WT + CQY forms stronger X-dimers than W2A, and also WT + CQY can still form S-dimers, whereas W2A cannot.

While our data demonstrates that endocytosis is the primary contributor to Pcad turnover, we cannot eliminate the contribution of Pcad diffusion towards increased junctional dynamics. Previous FRAP measurements on adherens junctions demonstrate that rapid endocytosis is the overwhelming contributor to dynamics in mature Ecad junctions[30]. In contrast, when adherens junctions are disrupted by chelating Ca$^{2+}$, Pcad diffusion increases significantly[30,34]. To decrease the effect of non-physiological diffusive contributions to Pcad dynamics, we performed FRAP experiments on cells that were cultured for at least 24 h to form mature junctions, and we performed all experiments in the presence of Ca$^{2+}$. Since endocytosis was shown to occur on the minute timescale[30,31], we monitored fluorescence recovery for ~3 min to capture rapid endocytosis events.

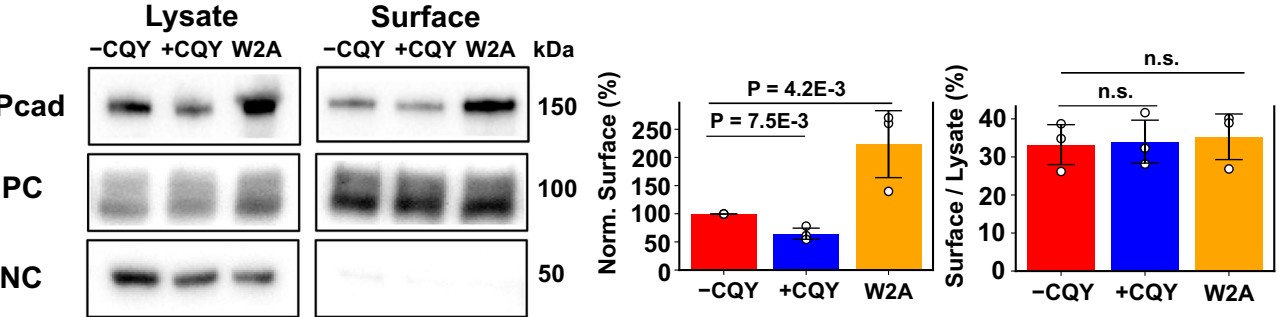

## d. Surface Biotinylation

Our manuscript reinforces the role of p120 as a primary regulator of cadherin dynamics on the cell surface. Unlike previous studies[12,13] that mutated the cadherin cytoplasmic regions to prevent p120 binding, we used an antibody targeting the ectodomains of Pcad. This approach promoted p120 phosphorylation and dissociation by trapping the Pcad ectodomains in an X-dimer conformation. Our results establish an outside-in link between cadherin extracellular binding conformations and intracellular signaling. Recent studies suggest that changes in the phosphorylation state of p120 regulate cell adhesion[18]. Specifically, antibodies 19A11 and 66E8, which strengthen Ecad S-dimers and enhance Ecad-mediated cell adhesion, were shown to

cause p120 de-phosphorylation in Colo-205 cells[17,18]. It is possible that the Ecad S-dimers signal p120 dephosphorylation and increase p120 association to Ecad, consequently, strengthening the cell adhesion mediated by Ecad. A recent computational study also proposes that p120 can stabilizes cadherin *cis*-interactions from the inside-out[35]. The binding/unbinding of other cytoplasmic proteins such as vinculin have also been recently shown to regulate the conversion of cadherin X-dimers to strand-swap dimers[21].

While we demonstrate that X-dimer formation phosphorylates p120, the molecular mechanism by which this occurs remains unresolved. Since several protein kinases such as Protein Kinase C-alpha

**Fig. 4 | CQY684 disrupts Pcad mediated cell-cell adhesion by promoting endocytosis. a** Dispase assay shows that the addition of 200 nM CQY684 Fab disrupts cell adhesion, and cell sheet monolayers become more fragmented. $N = 8$ for each condition, distributed across three independent biological repeats. **b** Cell aggregation assay shows that the addition of 200 nM CQY684 Fab disrupts cell adhesion in the A431 cells expressing WT Pcad. A431 cells expressing W2A Pcad mutants do not form aggregates in both presence and absence of CQY684. Measurements performed across three biological repeats. **c** Immunofluorescence confocal imaging shows that, in the presence of 200 nM CQY684, WT-Pcad colocalizes with the lysosomal marker LAMP1. Scale bar: 10 μm. Colocalization (yellow) of Pcad (green) and LAMP1 (red) is seen in the +CQY condition. $N = 163$ cells (WT − CQY), and 131 cells (WT + CQY) condition across three biological repeats. **d** Surface biotinylation assay. Left panel: Representative data. Cytoplasmic alpha-tubulin serves as negative control (NC) while membrane protein CD98 serves as positive control (PC). Similar total protein amounts (as measured with a DC protein assay) were loaded in each lane. Middle panel: Amount of cell surface Pcad compared to the −CQY condition. The amount of surface protein in the WT cells was reduced upon addition of 200 nM CQY684 due to increased internalization and lysosomal proteolysis of Pcad. W2A cells exhibited higher levels of surface Pcad compared to WT cells due to higher transfection efficiency. Right panel: Bar-plot of surface Pcad versus the total amount of Pcad (surface/lysate). Differences between the three conditions are non-significant. Surface/lysate Pcad ratio is similar in all conditions because the total amount of Pcad in the cell is reduced upon CQY binding, due to increased lysosomal proteolysis. Three independent repeats were performed in surface biotinylation assays. In all boxplots, the box represents the 25th and 75th percentiles with the median indicated and whiskers reach 1.5 times the IQR. Data points outside the whiskers are shown as outliers. Error bars in all bar plots are the standard deviations. Two-sided student $T$-test was used to evaluate the significance difference.

(PKCα)[36] and Casein Kinase I epsilon (CK1ε)[37] are known to change the phosphorylation state of p120, one possibility is that a protein kinases specifically recognizes the X-dimer conformation and phosphorylates p120. Alternatively, since p120 phosphorylation sites are adjacent to the p120 binding site, X-dimer formation may allosterically induce conformational changes on the Pcad cytoplasmic region, thereby allowing access to a protein kinase (Supplementary Fig. S13).

In summary, our study shows that CQY684 traps Pcad in an X-dimer conformation and stabilizes this adhesive structure. Formation of stable X-dimers leads to the phosphorylation of p120 and its dissociation from the cadherin cytoplasmic region, thereby promoting Pcad turnover and targeting Pcad to the lysosome. Our results suggest an outside-in signaling mechanism that can be exploited by anti-cadherin antibodies for intracellular drug delivery.

## Methods

### Purification of human WT, W2A, and K14E Pcad ectodomains and CQY684 Fab
The ectodomain of Pcad (1-654) was cloned with a C-terminal Avi-tag and 6XHis-tag and incorporated into pcDNA3.1(+) as previously described[14,15]. The mutant W2A and K14E Pcad were constructed using the Q5 site-directed mutagenesis kit (New England Biolab). The plasmids were transfected into Expi293 suspension cells with the Expi-Fectamine transfection kit (ThermoFisher Scientific). After 5 days of expression, the conditioned media containing Pcad ectodomains was collected, and a protease inhibitor tablet (ThermoFisher Scientific) was added to the media.

The WT and mutant Pcads were affinity purified with Ni-NTA agarose beads (Qiagen) in a gravitational column. The Ni-NTA beads were first washed with binding buffer (pH 7.5, 20 mM HEPES, 500 mM NaCl, 1 mM CaCl$_2$), followed by washing with biotinylation buffer (pH 7.5, 25 mM HEPES, 5 mM NaCl, 1 mM CaCl$_2$). The Ni-NTA-bound Pcads were then biotinylated using the BirA enzyme (BirA 500 kit, Avidity). The biotinylated proteins were eluted with elution buffer (pH 7.5, 500 mM NaCl, 1 mM CaCl$_2$, 200 mM imidazole). The eluted proteins were dialyzed to remove imidazole using storage buffer (pH 7.5, 10 mM Tris-HCl, 100 mM NaCl, 10 mM KCl, 2.5 mM CaCl$_2$). 20% Glycerol was added into the protein sample before the samples were flash-frozen in liquid nitrogen and stored at -80 °C.

The sequence of CQY684 Fab was previously described[19] and deposited in PDB (access code 6ZTR). CQY Fabs were recombinantly generated by cloning the variable regions of the heavy and light chains into human IgG Fab backbone by GenScript.

### Single-molecule AFM experiments
The single-molecule AFM experiments were performed using a previously described protocol[14,15,21]. Briefly, the cantilever (Hydra-2R-50N, AppNano) and glass coverslips (CS) were submerged in Piranha solution (25% H$_2$O$_2$, 75% H$_2$SO$_4$) overnight and washed with DI water twice.

The CS was also treated with 1 mM KOH and washed with DI water twice. The cantilever and CS were then washed with acetone and silanized with 2% 3-aminopropyltriethoxysilane (Millipore Sigma) in acetone for 30 min at room temperature. After that, the cantilever and CS were functionalized with 10% biotin-PEG-Succinimidyl Valerate (MW 5000, Laysan) and 90% mPEG-Succinimidyl Valerate (MW 5000, Laysan) in 100 mM NaHCO$_3$ and 600 mM K$_2$SO$_4$ for at least 4 h of incubation and then washed with DI water. Prior to the AFM experiments, the cantilever and CS were first blocked with 10 mg/mL BSA for 1 h at room temperature and then incubated with 0.1 mg/mL streptavidin (Invitrogen) for 30 min at room temperature. Finally, the functionalized cantilever and CS were incubated with 200 nM of purified proteins for 90 min.

All AFM experiments were carried out using an Agilent 5500 AFM system with a closed-loop scanner. The single-molecule force measurements were performed in a Tris buffer (pH 7.5, 10 mM Tris-HCl, 100 mM NaCl, 10 mM KCl, 2.5 mM CaCl$_2$). The spring constants were calculated using the thermal fluctuation method. To filter out the non-specific binding events, all recorded PEG stretching force curves were fitted to a worm-like-chain (WLC) model using the least-squares fitting method. The populations of unbinding forces were fitted using a Gaussian mixture model through binning based on the Freedman–Diaconis rule. Bayesian Information Criterion (BIC) was used to determine the optimal number of Gaussians in the unbinding force distributions.

### Bead aggregation assay
Biotinylated ectodomains of WT, W2A, and K14E Pcad at a concentration of 200 nM were mixed with Dynabeads™ MyOne™ Streptavidin C1 (Thermo Fisher Scientific) and incubated for 30 min in binding buffer (10 mM Tris-HCl, 150 mM NaCl, 0.05% Tween 20, 1 mg/ml BSA, pH 7.5) at 4 °C with gentle shaking at 90 RPM. Following incubation, the Dynabeads were washed and then transferred to either Calcium binding buffer (10 mM Tris-HCl, 150 mM NaCl, 5 mM CaCl$_2$, 0.05% Tween 20, 1 mg/ml BSA, pH 7.5) or CQY684 binding buffer (100 nM CQY684 Fab, 10 mM Tris-HCl, 150 mM NaCl, 5 mM CaCl$_2$, 0.05% Tween 20, 1 mg/ml BSA, pH 7.5) and vigorously rotated at room temperature for 2 h. After incubation, 10 μl of the sample was placed onto a sample slide and imaged using a 10X objective with an Olympus CX40 microscopy system. Analysis was performed using imageJ2 with a 30-pixel cutoff.

### Molecular Dynamics (MD) and Steered Molecular Dynamics (SMD) simulations
MD simulations were conducted on the FARM high-performance computing cluster at the University of California, Davis, using GROMACS 2022.3 as previously described[14,15]. The simulations utilized the OPLS-AA/L force field and the TIP4P water model with a 10 Å radius cut-off for Van der Waals and electrostatic interactions. Electrostatic

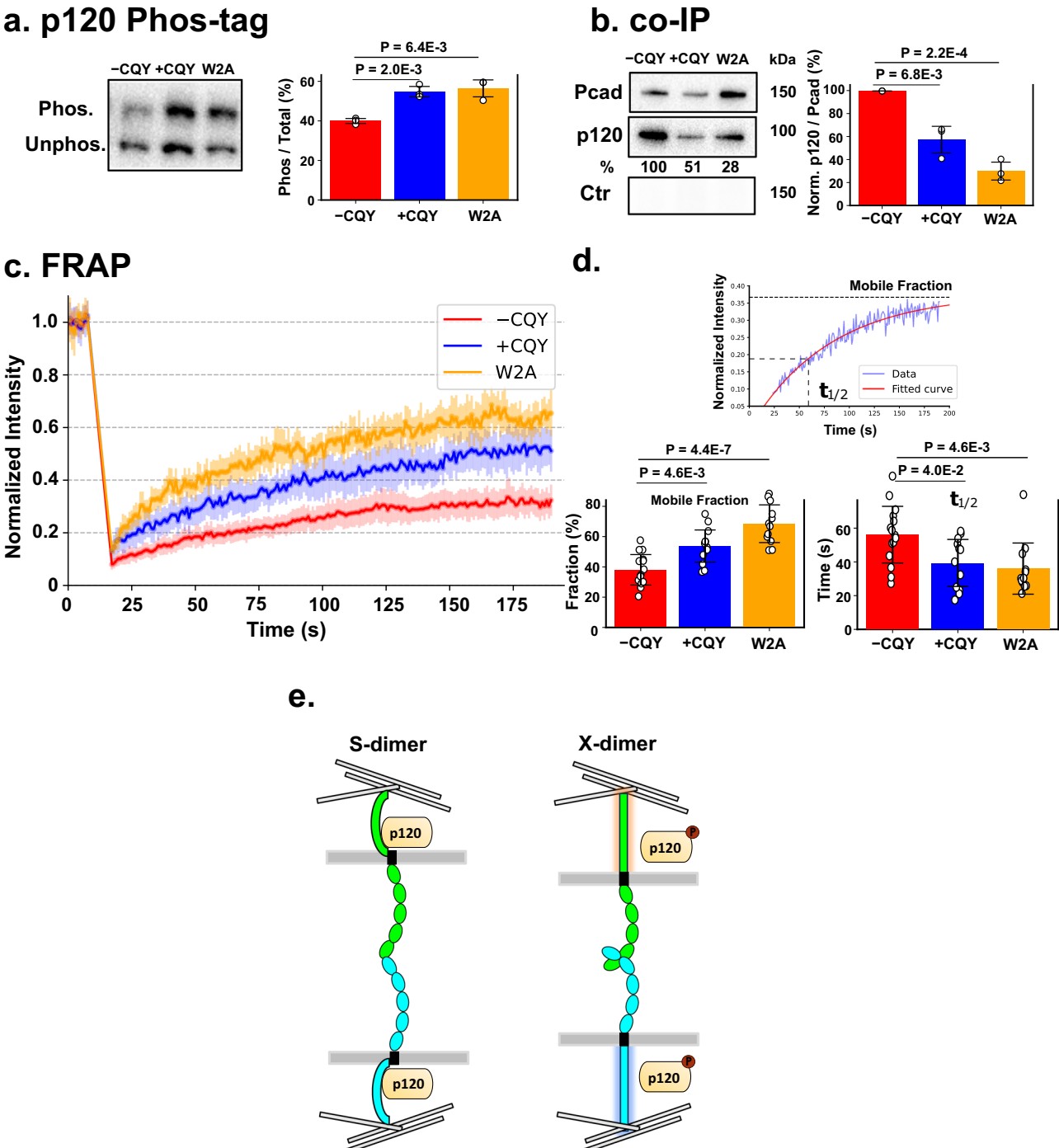

**Fig. 5 | X-dimer formation induces p120-catenin phosphorylation and dissociation. a** Phos-tag gel experiments show that p120-catenin gets more phosphorylated in X-dimers (i.e. either upon addition of 200 nM CQY684 or in the W2A mutants). Left panel: Representative p120 phos-tag gel data showing p120 exists in two phosphorylation states. The same amount of total protein (as measured with a DC protein assay) was loaded into each lane. Right panel: Bar-plot of the fraction of phosphorylated proteins in three replicates. **b** Co-immunoprecipitation (co-IP) experiments show that the amount of p120 association drops in X-dimers (i.e. either upon addition of 200 nM CQY684 or in the W2A mutants). Left panel: Representative co-IP data. The same amount of total protein (as measured with a DC protein assay) was loaded into each lane. No Pcad was detected when using an IgG control. Right panel: Bar-plot of the ratio of p120/Pcad measured in three replicates. All values are compared to the −CQY conditions. **c** FRAP experiments were performed on cell-cell junctions in WT − CQY (red line), WT + CQY (blue line), and W2A (orange line) conditions. Normalized, average fluorescence signal of Pcad tagged

with mCherry was measured. Error bars for each data point show 95% confidence intervals. $N = 20$ for each condition measured across three biological repeats. **d** Upper panel: Fluorescence recovery was fitted to a single exponential model. Bottom left: comparison of the mobile fraction across the three conditions. Bottom right: comparison of the half recovery time across the three conditions. Both fluorescence recovery rate and mobile fraction increased in +CQY and W2A conditions, indicating more dynamic cell-cell junctions. Only data with $R^2 > 7$ was included in analysis. Sample sizes are 17, 12, and 15 for −CQY, +CQY, and W2A conditions across three biological repeats. **e** Schematic of proposed model. S-dimers are bound to dephosphorylated p120 and are stable on the cell membrane. Formation of an X-dimer phosphorylates p120 which triggers p120 dissociation. This increases the turnover of X-dimers. Error bars are standard deviations unless otherwise specified. Two-sided student $T$-test was used to evaluate the significance difference.

energy was calculated using the particle mesh Ewald method with a 0.16 nm grid spacing. The EC1-2 Pcad X-dimer structures and CQY bound X-dimer complex structures were prepared by generating symmetry mates in PyMOL using the crystal structure PDB: 6ZTR. The structures were then prepared for simulations using PDBFixer.

At the start of the MD simulations, the Pcad X-dimer structures either bound or not bound to CQY684 were placed in the center of a dodecahedral box, ensuring every atom was at least 1 nm away from the boundary. The system was relaxed with energy minimization and stabilized with equilibration under isothermal-isochoric and isothermal-isobaric conditions using a modified Berendsen thermostat and Berendsen barostat. The box was filled with water molecules and neutralized with charged ions (100 mM NaCl, 4 mM KCl, and 2 mM CaCl$_2$).

After stabilization, a 60 ns MD simulation was performed with 2 fs integration steps, with the protein structure typically reaching equilibration after ~20 ns (Supplementary Fig. 6). Using the gmx rmsf module, the C-α RMSF of residues 1–100 in the EC1 domain was monitored from the backbone RMSD of the structure relative to the initial frame. To calculate the RMSF of the structure during the MD simulations, we used gmx rmsf module.

Simulations for p120 bound to the Ecad cytoplasmic region was constructed from the crystal structure PDB: 3L6X. The protein complex was similarly prepared as described above. However, only 20 ns MD simulations were performed.

The constant force SMD simulation were performed on the FARM high-performance computing cluster. Using the final frame from the MD simulation, we obtained an interacting Pcad protein structure and placed it in the center of a rectangular box, aligned with the box's longest axis, ensuring no atom was closer than 1 nm to the boundary (dimensions: 30 × 12 × 8 nm for the -CQY conditions; 30 × 15 × 15 nm for the +CQY condition). The system, containing ~38,000 atoms for the -CQY condition and ~88,000 atoms for the +CQY condition, was relaxed and equilibrated using the same method as in the MD simulation, but without the isothermal-isochoric step.

At the start of the SMD simulation, one C-terminus of the interacting Pcad protein pair was fixed while the other C-terminus at residue 213 was pulled along the longest axis of the box with a constant force of ~665 pN (400 kJ·mol$^{-1}$·nm$^{-1}$). To evaluate the stability of the structure throughout the SMD simulation, the gmx sasa module was used to calculate the ΔSASA (ΔSASA = SASA [protein A] + SASA [protein B] − SASA [protein A + protein B]), which estimated the change in the Pcad interfacial binding area.

### Generation of stable cell lines
WT-Pcad and W2A-Pcad full-length plasmids fused with mCherry at the C-terminus were introduced into A431 Ecad/Pcad double knockout cell line. Using the limiting dilution-culture method, a single clone of each rescue cell line was acquired. The monoclonal cell line was then expanded and re-selected using fluorescence-activated cell sorting (FACS). A single cell was again seeded onto collagen-treated 96-well plates, completing the generation of stable monoclonal A431 Pcad-mCherry rescued cells. The rescued cells were cultured in Gibco high glucose Dulbecco's Modified Eagle Medium (DMEM) containing 10% fetal bovine serum (FBS) and 1% penicillin-streptomycin.

### Cell aggregation assay
A431 Pcad-mCherry cells were detached using 0.05% trypsin diluted with calcium and magnesium-free solution (CMFS). To prevent the digestion of Pcad on the surface, 1 mM CaCl$_2$ was added to the trypsin buffer. After 30 min of trypsinization, the cells were suspended, and the reaction was stopped by adding suspension buffer (Life Science Ca-free DMEM with 10% Ca-free FBS) to recollect all suspended cells. The cells were washed again with suspension buffer and diluted to a concentration of 1 × 10$^5$ cells/ml. Then, 500 µL of suspended cells were added to each well of a 24-well plate with the supplementation of 2 mM CaCl$_2$ or 4 mM EGTA in the presence or absence of CQY684. The 24-well plates were pretreated with 0.5% BSA overnight at 4 °C to prevent cell attachment to the bottom of the plate and were washed three times with PBS prior to start of the experiment. Cells were incubated at 37 °C and 100 rpm for 2 h. Brightfield images were acquired with an Olympus CKX41 imaging system.

Analysis was performed using imageJ2. Images were first converted to 8 bit format, and pixels corresponding to single cells were picked (single cells correspond to threshold of 40 pixels). Only aggregates that contain >5 cells were used in the analysis.

### Monolayer dispersion assay
The cells were seeded in 12-well plates in the presence or absence of CQY Fab at a concentration of 1 × 10$^5$ cells/ml. After 48 h of incubation, the cell monolayer was washed twice with high-glucose DMEM to remove any floating cells and then incubated with 1.2 U/ml Dispase II (Millipore Sigma) in PBS with 1 mM CaCl$_2$ for 1 h at 37 °C. After detachment, the lifted cell monolayer was placed on an orbital shaker and subjected to a shaking force of 200 rpm for 40 min. The fragmentation of the cell monolayer was imaged and analyzed using a Leica confocal imaging system.

### Immunofluorescence confocal imaging
Cells were plated on a Bovine Collagen I (R&D Systems) coated glass coverslip and cultured in the presence or absence of CQY684 Fab until reaching 70% confluence. The cultured cells were fixed with an ice-cold 1:1 acetone/methanol mixture for 10 min at 4 °C. After fixation, the cells were washed three times with PBS and blocked with 5% BSA buffer. The blocked samples were then incubated with a 1:100 dilution of primary antibody for 1 h, followed by a 1:500 dilution of secondary antibody for 30 min at room temperature. Images used to quantify internalization were captured with a Leica confocal microscopy system. Colocalization analysis for Pearson's coefficients was carried out using ImageJ2 module colo2 with Costes threshold regression method; PSF set to 3 and Costes randomization set to 50. For the colocalization analysis, we specifically acquired images in a single z-plane. To allow comparison, all images were acquired at the same excitation laser intensity, pinhole size, and smart gains. The pixel size for each image was adjusted to ~20 nm x 20 nm.

### Fluorescence Recovery After Photobleaching (FRAP)
To prepare samples for FRAP experiments, cells were seeded onto Cellvis 35 mm dishes coated with Cell-Tak (Fisher Scientific) and cultured for at least 24 h. During incubation, 200 nM CQY684 was added to the A431 cells expressing WT Pcad for the '+CQY' condition. Subsequently, cells were transferred to a Leica Stellaris 5 confocal microscopy system equipped with FRAP module. Imaging was performed using a 63X oil immersion objective with a 1 Airy unit pinhole. mCherry fused to Pcad was excited using a 568 nm argon laser, with 2% laser intensity for pre- and post-bleaching imaging, and 30% laser intensity during bleaching. The FRAP protocol consisted of acquiring 10 pre-bleach frames, followed by 10 frames during bleaching, and 200 post-bleach frames (863 ms exposure per frame). Mean integrated fluorescence intensity within each region of interest (ROI) was calculated and normalized to the pre-bleaching value. Background measurements from the same ROI were subtracted during calculations. The recovery phase of the fluorescence signal was fitted to a single exponential equation: $y = A(1 - e^{-k(t-t_0)})$. Only R$^2$ values above 0.7 were used in subsequent analysis.

### Co-Immunoprecipitation
To prepare the cell lysate for co-immunoprecipitation (co-IP), cells were seeded in a T75 tissue culture flask (Thermo Fisher Scientific) in the presence or absence of CQY684 Fab. After 48 h of incubation, the

cells were washed three times with PBS and scraped off with 1 ml PBS. The cells were then resuspended in M2 lysis buffer (pH 7.5, 50 mM Tris-HCl, 150 mM NaCl, 1% SDS, 1% Triton-X 100) with 1% protease inhibitor cocktail (Sigma-Aldrich) and 0.1% Benzonase (Millipore Sigma). The resuspended cell solution was flash-frozen in liquid nitrogen and incubated on ice for 30 min. The cell lysates were then sonicated for two 1 min cycles to ensure full release of proteins. After the lysis process, the cell lysates were centrifuged at 13,000 rpm for 10 min at 4 °C to remove cell debris, and the protein concentration was determined using the Bio-Rad DC protein assay with BSA as the standard.

Prior to the co-IP experiments, magnetic protein G beads (Thermo Fisher Scientific) were incubated with anti-mCherry rabbit mAb #43590 (Cell Signaling Technology) at room temperature for 10 min. Then, 500 μl of cell lysate was added to the antibody-bound beads and incubated for 30 min with rotation at room temperature. The beads were washed three times with washing buffer (Thermo Fisher Scientific). The antigen proteins were eluted by incubating the beads with elution buffer (Thermo Fisher Scientific) for 2 min with rotation at room temperature.

### Surface biotinylation assays
Surface biotinylation was performed as previously described[38–40]. Briefly, a confluent monolayer of cells formed in the presence or absence of CQY Fab was washed three times with cold PBS. Cell surface proteins were incubated for 10 min at 4 °C with EZ-link Sulfo-NHS-SS-Biotin (Thermo Fisher Scientific). Then, 500 μL of quench buffer (Thermo Fisher Scientific) was added to stop the reaction, and the cells were recollected. The cells were lysed with M2 lysis buffer, and the lysates were incubated with NeutrAvidin agarose (Thermo Fisher Scientific) for 60 min at room temperature with end-over-end mixing using a rotator. The agarose beads were then washed three times with washing buffer (Thermo Fisher Scientific) and eluted by incubating the beads with SDS-PAGE sample buffer containing 50 mM DTT. The amount of protein in the cell lysate and elution was measured from western blots.

### Phos-tag gel
To investigate the phosphorylation levels of target proteins, cell lysates from cells treated with or without CQY Fab were supplemented with Pierce™ Phosphatase Inhibitor at the recommended concentration. Proteins at different phosphorylation levels were separated using a SuperSep™ Phos-tag gel (50 μmol/L, 7.5%). Protein amounts were detected by western blot and analyzed with ImageJ.

### Western blots
Purified human WT and mutant W2A/K14E Pcad ectodomains, cell lysates, and eluted proteins from co-IP and surface biotinylation experiments were prepared by boiling at 95 °C for 5 min in SDS-PAGE sample buffer (Bio-Rad, 90% 4X Laemmli Sample Buffer + 10% 2-Mercaptoethanol). The denatured samples were run on 4–15% polyacrylamide gels (Mini-PROTEAN TGX Precast Protein Gels, Bio-Rad) at 200 V for 30 min and then transferred to a PVDF membrane (Bio-Rad) at 200 mA for 1 h on ice. The membrane was then blocked with 5% blocking buffer (PBS + 0.1% Tween 20 + 5% blotting-grade blocker) for 1 h and washed three times with PBST. After a 1 h incubation with a 1:1000 dilution of the primary antibody and a 1:5000 dilution of the secondary antibody, the protein was detected using WesternBright ECL HRP substrate (Advansta).

### Statistical analysis
Statistical analysis was performed using Python and Excel. Since this research primarily involved pairwise comparisons, the results were analyzed using a two-sided Student's $t$-test. $P$-values are presented in the figures.

### Reporting summary
Further information on research design is available in the Nature Portfolio Reporting Summary linked to this article.

## Data availability
All processed data have been made available in the manuscript and supporting information. Simulation input files and a coordinate file of the final output have been deposited in a public repository with the identifier https://doi.org/10.5281/zenodo.13787595. Source data are provided with this paper.

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

## Acknowledgements

This research was supported by the National Institute of General Medical Sciences of the National Institutes of Health (R01GM133880) to SS. We thank Prof. Sergey Troyanovsky (Northwestern University) for sharing the A431 Ecad/Pcad double knockout cells. FACS was performed at the Flow Cytometry Shared Resource funded by the UC Davis Comprehensive Cancer Center Support Grant (CCSG) awarded by the National Cancer Institute (NCI P30CA093373).

## Author contributions

B.X., S.X., and S.S. designed the research. B.X., and S.X. performed the research and analyzed the data. S.S. supervised the research. B.X., S.X. and S.S. wrote the paper.

## Competing interests

The authors declare no competing interests.
