## [Peer Review file · Nature Communications]

Outside-in engineering of cadherin endocytosis using a conformation strengthening antibody.

Corresponding Author: Professor Sanjeevi Sivasankar

Version 0:

Reviewer comments:

Reviewer #1

(Remarks to the Author)

The authors present an interesting and insightful study on a novel outside-in signaling mechanism involving cadherins, specifically focusing on the Pcad X-dimer conformation. This work highlights important aspects of cadherin dynamics and their potential implications for therapeutic strategies. The study demonstrates that the monoclonal antibody (Mab) CQY684 binds to the Pcad EC1 domain, stabilizing the X-dimer conformation, which leads to the dissociation of p120-catenin from the cadherin cytoplasmic region. This, in turn, triggers Pcad internalization and lysosomal targeting, offering possibilities for intracellular drug delivery. Notably, CQY684 selectively stabilizes the X-dimer without affecting the S-dimer conformation.

In addition, the authors propose a model that compares stretched versus compressed intercellular junctions and links this with the phosphorylation-induced inactivation of p120 proteins. While the work is certainly interesting and provides valuable insights, I feel that some of the conclusions may be somewhat speculative without stronger experimental evidence. Including additional experimental support would further strengthen the article and bolster its impact.

Following are the major concerns –

A. Page 5: Last Paragraph:

1. Why are the emergent salt-bridge interactions between cadherins in the presence of CQY considered stochastic (exclusively), rather than the salt-bridge interactions typically responsible for X-dimerization?
2. Based on the same reasoning, wouldn't one also expect to observe a peak at ~25 pN when CQY does not stabilize the X-dimer in a stochastic manner?
3. It is intriguing that a single salt-bridge enhances the overall interaction strength by ~20 pN, which is comparable to the X-dimer interaction strength (composed of many salt-bridge and non-covalent interactions) without CQY. Could the authors provide an explanation for such a significant increase in unbinding force? Simulation results might offer valuable quantitative insights into this.
4. Do the weightings in the bimodal distributions change with different pulling speeds? Could the authors provide comments on this?

B. Page 6: 1st Paragraph: Line – 'For Pcad K14E mutants, which can only form S-dimers, single Gaussian distributions....'
This statement may not be entirely accurate without an estimate of the binding affinity. The current findings only suggest that, at the solution concentration of CQY used in this experiment, no experimental evidence of additional stability was observed. To confirm this, the authors could consider conducting experiments at higher concentrations. Alternatively, they could provide justification based on the relationship between solution-based affinity and 2D affinity to support their conclusions.

C. Page 6: 1st Paragraph: Line : 'Since CQY684 only strengthens X-dimers and does not affect S-dimers, we concluded that the observed....'

The results suggest that the binding of CQY to the X-dimer occurs more quickly than the conversion of the X-dimer to the S-dimer. This is a particularly interesting and unexpected observation, as one would conventionally expect the opposite, given that the X-dimer is typically an intermediate and is known to rapidly convert to the S-dimer within the timescale of force

spectroscopy measurements. This rapid conversion likely explains why the existence of the X-dimer was not observed by the authors during pulling experiments with wild-type cadherins. Therefore, further evaluation is needed. I have the following suggestions:

(1) The authors may consider performing kinetic measurements to quantify the rate differences between the X-to-S dimerization rate and the binding rate of CQY to the X-dimer.

(2) If direct kinetic measurements are challenging, the authors could vary the concentration of CQY to modulate the binding rate and identify a limiting condition where the X-to-S transition would dominate.

D. Page 7: 1st Paragraph: 1st Line: A slow on-rate may indeed slow down the aggregation process, but one would still expect to observe some degree of aggregation. How long did the authors run the measurements? Additionally, is there a way to provide a quantitative estimate for this observation? It would be helpful to clarify whether the duration of the experiment was sufficient to capture potential aggregation events despite the slow on-rate.

E. Page 11: 2nd Para: 'To understand why CQY684 disrupts cell adhesion while strengthening the Pcad X-dimer conformation, we hypothesized....' –

(1) What is the rationale behind such a hypothesis, especially considering that the X-dimer with CQY shows increased interaction strength even greater than S-dimers?

(2) What specifically signals increased turnover in this context?

(3) Turnover refers to the dynamic balance between protein expression and degradation, and typically, a higher turnover rate should maintain homeostasis. What causes this balance to be disturbed in this case? Can the authors provide more insights into the factors driving the disruption in this cadherin turnover, particularly in the presence of CQY?

F. Page 11: 2nd Para: last Line: Is it inducing more turnover or inducing more protein cleavage for internalization?

G. What is DC assay?

H. Finally, it would be beneficial if authors provide additional experimental supports for the model of stretched vs relaxed cell-cell junction;

(1) Strength of cell-cell junction with/without CQY while arresting/blocking internalization.

There are certain minor comments related to referencing and the comments are directly made on the manuscript.

Reviewer #2

(Remarks to the Author)

This study by Xie et al combines single molecule experiments, simulations, and cell-based experiments to determine the mechanism of action of the PCadherin binding antibody CQY. This multi-faceted multi-scaled approach is a strong way to address this important complex biophysical and cell biological question. Understanding this mechanism is important for both drug delivery and basic cell biology, making it of great interest. However the results are not supported as key controls and orthogonal validations are lacking and details of some experiments and statistical tests are not well described. Statistical methods are not well described. At times the results jump, referencing experiments that have not yet been described which is difficult to follow. These were the main

A key conclusion of this study is that CQY binding traps Pcad in an X-dimer, leading to dissociation of p120 which promotes PCad endocytosis and degradation.

The data suggesting less PCad on the surface is Fig4d. This experiment lacks controls, at a minimum positive and negative controls for the assay as well as a surface protein not impacted by CQY should also be probed. The elute/lysate is confusing - in the text 40% is mentioned but this plot shows 100%. That there is not a change in ratio between total Pcad and surface in the 2 treatment groups is surprising seeing the microscopy. Along with the controls, an orthogonal experiment demonstrating there is less PCad on the cell membrane would strengthen the conclusions.

The data suggesting there is less p120 binding to pcad is presented in Fig4e. It is not clear if the reduction in p120 binding reflects the smaller proportion of pcad present on the cell membrane or less p120 binding for the pcad that is at the membrane. The control box is completely empty, this cannot be interpreted. The authors refer to the elute/lysate in Fig4e as justification for the conclusion. However this data needs controls (see above) and seems to counter the imaging which shows significantly more internalized pcad in the +CQY cells. This should be validated with orthogonal experiments such as p120 staining. Can biochemistry be used to pull down only CQY bound pcad and probe p120 binding?

The authors should show that at the timepoints of their experiments CQY is still bound to the cell surface.

Critically the concentration of CQY is not reported for multiple experiments in the legend or Materials and Methods. When p values are reported, the statistical test used is not included and often it is challenging to determine what was considered "n". Some p values seem extremely small.

Please mention any implications in the modeling for using the PCad Xdimer +CQY but simply removing the CQY Fabs (P7L18-19)? Could the antibody binding change the X-dimer conformation?

Described below are some additional questions.

Figure 1 What is considered the “n” in these experiment?

Figure 4b what is “n”? And what is the y axis (cell counts) in the context of this experiment, I understood it to be an aggregation experiment??

Fig 4c Co-localization. Is this a single z plane? The LAMP1 staining does not look the same between the +/- CQY conditions. Indeed, there is hardly any LAMP-1 in the control condition.

Are the images acquired and scaled in the same way so intensities are comparable? Additionally, the images appear to be scaled differently looking at the 10 um scale bars. What do the “n” in the figure legend correspond to? Number of images acquired?

Reviewer #3

(Remarks to the Author)

This work used a combination of biophysical, computational, biochemical and cell biological methods to demonstrate that an antibody, CQY684, can trap P-cadherin in its X-dimer conformation, which can trigger P120 dissociation and lead to the endocytosis of P-cadherin. These discoveries offer a previously unknown mechanism of outside-in signaling and provide new insights to the drug delivery. The overall research is well designed, and the results are highly interesting. I recommend the acceptance of the manuscript to Nature Communications after the authors provide the following justifications.

1. the construction of the complex formed by P-cadherin X-dimer and antibody needs more details. The original crystal structure only contains cadherin's first two N-terminal domains and the antibody. Is it possible that the antibody could also bind to the cadherin in its S-dimer configuration?
2. This study doesn't mention another important interaction, the cis-interaction, which is critical to the formation of junction. The antibody binding interface overlaps with the cis binding interface, based on the observation. Could the author add justification to discuss how cis interaction could be affected with the presence of antibodies?
3. In addition to the outside-in signaling, the inside-out signaling has also recently been proposed to the cadherin-p120 system, in which the binding of p120 to the cytoplasmic region of cadherin could change the configuration of its extracellular domains, changing its intercellular trans or cis interactions, and thus regulating cell adhesion. The authors should mention this in either introduction or discussion.
4. The SMD simulation of Fig 5b does not provide direct evidence to how x-dimer formation leads to dephosphorylation of p120, and it's kind of straightforward that proteins would dissociate if being pulled apart. Therefore, the authors could either remove this simulation part or put it in the supplementary material.

Version 1:

Reviewer comments:

Reviewer #1

(Remarks to the Author)

Authors have meticulously addressed all the concerns. The manuscript is now ready for publications without any changes.

Reviewer #3

(Remarks to the Author)

The revised manuscript is much better, the authors addressed all my comments, I am satisfied. I think the current version of the paper is ready for publication on Nature Communications!

Response to Reviewers

We appreciate all the reviewers' comments and suggestions. We believe that incorporating the reviewer's feedback has significantly strengthened our manuscript.

First, we would like to highlight the major changes we incorporated.

1. We added additional SMD analysis, mapping interactions during X-dimer dissociation in Figure 3d. This result provides further insights into how the formation of a single salt bridge significantly strengthens X-dimer interactions.
2. We conducted Phos-tag gel experiments showing increased phosphorylation of p120 upon the addition of CQY684 or with the W2A mutation (Figure 5a). This finding supports our model, where formation of Pcad X-dimers trigger p120 phosphorylation, leading to p120 dissociation and increased cadherin turnover.
3. We also performed additional analysis and control experiments requested by the reviewers (Figure 4d and Supplementary Figures S3, S4, S8, and S9)

We also reorganized Figures 4 and 5, moving 'less significant' results to the supplementary material, and adjusted the color scheme in Figure 2 for consistency.

Below is our point-by-point response to the reviewers. The original reviewer comments are highlighted in *blue*, and our response is in black.

Reviewer #1 (Remarks to the Author)

The authors present an interesting and insightful study on a novel outside-in signaling mechanism involving cadherins, specifically focusing on the Pcad X-dimer conformation. This work highlights important aspects of cadherin dynamics and their potential implications for therapeutic strategies. The study demonstrates that the monoclonal antibody (Mab) CQY684 binds to the Pcad EC1 domain, stabilizing the X-dimer conformation, which leads to the dissociation of p120-catenin from the cadherin cytoplasmic region. This, in turn, triggers Pcad internalization and lysosomal targeting, offering possibilities for intracellular drug delivery. Notably, CQY684 selectively stabilizes the X-dimer without affecting the S-dimer conformation.

In addition, the authors propose a model that compares stretched versus compressed intercellular junctions and links this with the phosphorylation-induced inactivation of p120 proteins. While the work is certainly interesting and provides valuable insights, I feel that some of the conclusions may be somewhat speculative without stronger experimental evidence. Including additional experimental support would further strengthen the article and bolster its impact.

We thank the reviewer for their positive feedback.

Following are the major concerns –

A. Page 5: Last Paragraph:

1. Why are the emergent salt-bridge interactions between cadherins in the presence of CQY considered stochastic (exclusively), rather than the salt-bridge interactions typically responsible for X-dimerization?

Response 1: At the outset, we would like to point out that there are **no salt bridge interactions responsible for X-dimerization in the absence of CQY**. On the contrary, X-dimerization is exclusively mediated by hydrogen bonds¹. When CQY684 binds to the X-dimer, a salt-bridge interaction (105LYS:199ASP) is stochastically observed due to Pcad conformational changes induced by CQY684 binding. During our MD simulations, we analyzed the electrostatic interactions (i.e. salt-bridges and hydrogen bonds) between Pcad in an X-dimer, both with and without CQY684, as shown in Figure R1 and **Supplementary Figure S7**. Without CQY684, the X-dimer is stabilized by ~6 'persistent' hydrogen bonds (defined as interactions lasting for more than 40% of the total simulation time). However, with CQY684 bound, we observed 5 persistent hydrogen bonds and 1 persistent salt bridge, indicating a change in the X-dimer binding conformation. The salt bridge between 105LYS and 199ASP was absent or non-persistent in the -CQY conditions, and only observed around 65% of the time in the +CQY conditions, indicating its stochastic nature. As shown in **Figure 2e**, this salt bridge is formed consistently only in sets 3–5 under +CQY conditions, demonstrating stronger interactions, while it remains absent in sets 1–2, further highlighting its stochastic nature.

In general, the stochastic behavior of a salt bridge may result from high salt concentration in solvent, and our single molecule experiments and simulations are performed in 100mM NaCl, which is similar to physiological high salt conditions. Both the simulations and single molecule experiments demonstrate bimodal force distributions with X-dimers and WT Pcad, which is also consistent with the salt bridge formation strengthening the X-dimer. We further elaborate on how a single salt bridge can strengthen X-dimer interactions in **response 3**.

We describe the change in electrostatic interactions induced by the binding of CQY684 on **lines 170-179**.

Figure R1. Electrostatic interactions between Pcad in an X-dimer change upon interaction with CQY684. Bar plots of all the electrostatic interactions, which include salt bridges and hydrogen bonds, that are persistent over 40% of the total simulation time. Salt bridge interactions are highlighted in red and hydrogen bonds interaction are colored in blue. The addition of CQY684 introduces a novel salt bridge 105LYS:199ASP.

2. Based on the same reasoning, wouldn't one also expect to observe a peak at ~25 pN when CQY does not stabilize the X-dimer in a stochastic manner?

Response 2: Both our simulations and AFM data demonstrate that Pcad X-dimers are strengthened upon binding of CQY even when the 105LYS:199ASP salt bridge does not form. This is because CQY stabilizes the X-dimer by binding-to and reinforcing three loops of Pcad, which in turn stabilize the adjacent X-dimer binding interface. As shown by our new data in response 3, formation of the 105LYS:199ASP salt bridge further strengthens the X-dimer.

From our MD and SMD results (**Figures 2 and 3**), even when the 105LYS:199ASP salt bridge does not form in sets 1 and 2 of the +CQY condition, the simulations show stronger interactions compared to all simulations without CQY: the dissociation time under constant pulling force in the absence of the salt-bridge is ~3500 ps compared to ~1200 ps in all simulations without CQY (**Figure 3c**). When the salt bridge forms (in sets 3–5), the interactions are even stronger, with dissociation times of ~7000 ps, indicating significantly enhanced stability. Thus, regardless of whether the salt bridge forms, CQY strengthens the Pcad X-dimer, meaning we would not expect to observe a force peak of ~25 pN in the presence of CQY. The binding of CQY stabilizes the Pcad X-dimer which is further strengthened when the 105LYS:199ASP salt bridge forms.

We now mention this on **lines 110-114** and describe it in mechanistic detail on **lines 202-220**.

3. It is intriguing that a single salt-bridge enhances the overall interaction strength by ~20 pN, which is comparable to the X-dimer interaction strength (composed of many salt-bridge and non-covalent interactions) without CQY. Could the authors provide an explanation for such a significant increase in unbinding force? Simulation results might offer valuable quantitative insights into this.

Response 3: We thank the reviewer for this suggestion. We would again like to begin by highlighting that X-dimerization in the absence of CQY is exclusively mediated by hydrogen bonds without salt-bridge formation. The formation of the 105LYS:199ASP salt-bridge is only induced upon CQY binding.

A salt bridge is generally considered a much stronger interaction (roughly 1.5 to 3 times) compared with a hydrogen bonds². To quantitatively test how formation of a single salt bridge could affect X-dimer dissociation, we characterized changes in electrostatic interactions (hydrogen bonds and salt bridges) during SMD simulations (Figure R2). In the presence of CQY but absence of salt bridge (+CQY -salt bridge; i.e. sets 1 and 2), the 105LYS:199ASP salt bridge briefly formed at the beginning of the SMD and ruptured within ~0.3 ns. Consequently, the remaining hydrogen bonds ruptured and the average dissociation time for X-dimers in the +CQY -salt bridge condition was ~3ns. In contrast, in the presence of CQY and salt bridge (+CQY +salt bridge; i.e. sets 3-5), the salt bridge lasted for ~6ns, which significantly enhanced the lifetime of the remaining hydrogen bonds and strengthened the X-dimer. In the +CQY +salt bridge condition, the average dissociation time of the X-dimer was ~7ns. The only major difference in terms of electrostatic interactions between “+CQY -salt bridge” versus “+CQY +salt bridge” conditions is the stability of the 105LYS:199ASP salt bridge and its effect on strengthening the hydrogen bonds at the X-dimer interface.

We now describe this on lines 210-220 and show the data in Figure 3d.

Figure R2. Stable 105Lys:199Asp salt bridge formation significantly strengthens X-dimer interactions. Violin plots of individual electrostatic interactions (salt bridges and hydrogen bonds) during SMD simulations under three conditions: in the presence of CQY684 without/with the formation of a persistent 105Lys:199Asp salt bridge (“+CQY -salt bridge” / “+CQY +salt bridge”), and in the absence of CQY684 (“-CQY”). For clarity, only electrostatic interactions lasting over 0.2 ns are shown. No significant differences were observed between “+CQY -salt bridge” and “+CQY +salt bridge,” except for the duration of the 105Lys:199Asp interaction (highlighted in magenta), indicating that the persistent 105Lys:199Asp salt bridge substantially strengthens X-dimer interactions.

4. Do the weightings in the bimodal distributions change with different pulling speeds? Could the authors provide comments on this?

Response 4: We thank the reviewer for raising this important point. Based on the reviewer’s comment, we performed new sets of AFM experiments with Pcad W2A in the presence/absence of CQY684 using a higher pulling speed of 5 $\mu\text{m/s}$ (compared to our original pulling velocity of 1 $\mu\text{m/s}$) (Figure R3). Consistent with the slower pulling velocity results, unbinding force histograms at the higher pulling speed was also best fit to a single Gaussian distribution in the absence of CQY684, and bimodal Gaussian distribution in the presence of CQY684 (Figure R3).

This result is now displayed in Supplementary Figure S3 and discussed on lines 106-109.

Figure R3. Pcad W2A AFM experiments performed at a higher pulling velocity (5 μ m/s) and the corresponding Bayesian Information Criterion (BIC) test. Histograms of Pcad W2A unbinding forces (a) without or (b) with CQY684 Fab. Unbinding force distribution was best described by a single Gaussian in the -CQY condition, and bimodal Gaussian distribution in the +CQY condition.

B. Page 6: 1st Paragraph: Line – ‘For Pcad K14E mutants, which can only form S-dimers, single Gaussian distributions....’

This statement may not be entirely accurate without an estimate of the binding affinity. The current findings only suggest that, at the solution concentration of CQY used in this experiment, no experimental evidence of additional stability was observed. To confirm this, the authors could consider conducting experiments at higher concentrations. Alternatively, they could provide justification based on the relationship between solution-based affinity and 2D affinity to support their conclusions.

Response 5: As suggested by the reviewer, we performed new AFM experiments with Pcad K14E in the presence of a much higher, 500 nM concentration of CQY684 compared to the 40 nM CQY684 concentration used in the original experiment. As seen in figure R4, the unbinding

force histogram measured in the presence of 500 nM CQY was similar to both the unbinding force histogram measured in the absence of antibody and the unbinding force histogram measured in the presence of 40 nM CQY. Under all experimental conditions, the histograms were best fit to a single Gaussian (Figure R4).

We have presented this results in **Supplementary Figure S4** and discussed on **lines 118-123**.

K14E +CQY 500 nM

Figure R4. AFM experiments with the K14E mutant measured at a higher CQY684 concentration and the corresponding Bayesian Information Criterion (BIC) test. Histograms (left panel) of Pcad K14E unbinding forces in the presence of 500 nM CQY684 Fab are best fit to a single Gaussian distribution (right panel).

C. Page 6: 1st Paragraph: Line :‘Since CQY684 only strengthens X-dimers and does not affect S-dimers, we concluded that the observed.....’

The results suggest that the binding of CQY to the X-dimer occurs more quickly than the conversion of the X-dimer to the S-dimer. This is a particularly interesting and unexpected observation, as one would conventionally expect the opposite, given that the X-dimer is typically an intermediate and is known to rapidly convert to the S-dimer within the timescale of force spectroscopy measurements. This rapid conversion likely explains why the existence of the X-dimer was not observed by the authors during pulling experiments with wild-type cadherins. Therefore, further evaluation is needed. I have the following suggestions:

(1) The authors may consider performing kinetic measurements to quantify the rate differences between the X-to-S dimerization rate and the binding rate of CQY to the X-dimer.

(2) If direct kinetic measurements are challenging, the authors could vary the concentration of CQY to modulate the binding rate and identify a limiting condition where the X-to-S transition would dominate.

Response 6: We thank the reviewer for giving us this opportunity to clarify aspects of our experimental design. In our AFM experiments, the coverslip and AFM tips were functionalized with Pcad monomers and incubated in a buffer containing an excess of CQY684 Fab **before** the

tip and coverslip were brought into contact. **Consequently, CQY684 was bound to the Pcad monomers before they formed *trans* dimers.**

The reviewer correctly noted that the X-dimer typically acts as an intermediate in the formation of S-dimers. It is widely believed that S-dimer formation first requires X-dimers to form³, although recent cryoEM⁴, high-speed AFM⁵, and NMR⁶ experiments suggest that X-dimers may be a 'standalone' adhesive structure. Regardless of the exact dimerization pathway, our simulation results (**Figure 3a and b**) demonstrate that CQY684 stabilizes the X-dimer, potentially trapping Pcad interactions in this state before the transition to S-dimers occurs. Therefore, our AFM data provides experimental evidence supporting this mechanism. We clarify this on **lines 191-196**.

D. Page 7: 1st Paragraph: 1st Line: A slow on-rate may indeed slow down the aggregation process, but one would still expect to observe some degree of aggregation. How long did the authors run the measurements? Additionally, is there a way to provide a quantitative estimate for this observation? It would be helpful to clarify whether the duration of the experiment was sufficient to capture potential aggregation events despite the slow on-rate.

Response 7: Our bead aggregation experiments lasted for approximately 2 hours and importantly, **our findings are consistent with previous studies using Ecad K14E, where no bead aggregation was observed within the same time frame**³. The reviewer is correct in suggesting that the duration of the experiment may not have been sufficient to capture potential aggregation events. We now clarify on **lines 142-144** that the time window of our experiments may not be long enough to observe K14E bead aggregation.

However, it remains unclear how long an experiment would need to run to detect K14E aggregation. If aggregation were to occur after a significantly longer duration, it could potentially result from protein degradation or the aggregation of unfolded protein. So, this phenomenon requires additional study, and is not the major focus of our paper.

E. Page 11: 2nd Para: 'To understand why CQY684 disrupts cell adhesion while strengthening the Pcad X-dimer conformation, we hypothesized....' –

(1) What is the rationale behind such a hypothesis, especially considering that the X-dimer with CQY shows increased interaction strength even greater than S-dimers?

Response 8: The rationale comes from a previous study⁷ showing that the X-dimers facilitates cadherin turnover. Even though the X-dimer bound to CQY interacts more strongly than S-dimers in cell-free experiments, cell-cell adhesion is disrupted by the addition of CQY suggesting that the CQY-X-dimer complex may get internalized. We have added this reference and described this rationale on **lines 256-258**.

(2) What specifically signals increased turnover in this context?

Response 9: We thank the reviewer for raising this important point. We have now assayed the phosphorylation state of p120 using phos-tag gel experiments (Figure R5). This data shows that p120 gets phosphorylated in the X-dimer conformation (i.e. either upon CQY binding or with the

W2A mutation). Numerous previous studies demonstrate that phosphorylation of p120-catenin results in dissociation of p120 from the Pcad cytoplasmic region^{8,9}.

These results have been included in **Figure 5a** and described on **lines 289-296**.

Figure R5. Phos-tag gel experiments show that p120-catenin is phosphorylated by the addition of CQY684 or in the W2A mutant. Left panel: Representative p120 phos-tag gel data showing p120 exists in two phosphorylation states. The same amount of total protein (measured with a DC protein assay) was loaded into each lane. The amount of phosphorylated (upper bands) and unphosphorylated (lower bands) p120 were determined by measuring western blots intensity. Right panel: Bar-plot of the fraction of phosphorylated proteins in three replicates.

(3) Turnover refers to the dynamic balance between protein expression and degradation, and typically, a higher turnover rate should maintain homeostasis. What causes this balance to be disturbed in this case? Can the authors provide more insights into the factors driving the disruption in this cadherin turnover, particularly in the presence of CQY?

Response 10: As discussed in Response 9, we believe that the phosphorylation of p120 disrupts Pcad homeostasis.

F. Page 11: 2nd Para: last Line: Is it inducing more turnover or inducing more protein cleavage for internalization?

Response 11: Previous studies indicate that phosphorylation and dissociation of p120 triggers increased cadherin turnover^{10,11}. In our study, we observed both p120 phosphorylation and dissociation when Pcad was trapped in an X-dimer upon CQY binding, which supports the hypothesis that CQY684 binding leads to greater Pcad turnover. However, cadherin cleavage, which is a prerequisite for lysosomal degradation, appears to be independent of p120 phosphorylation. Cells expressing the W2A mutant which are also trapped in an X-dimer conformation showed reduced Pcad cleavage despite p120 phosphorylation. Consequently, Pcad-W2A does not colocalize with the lysosome. In contrast, A431 cells expressing WT Pcad show substantial cadherin cleavage regardless of CQY treatment. Thus, it is unlikely that CQY684 directly triggers increased cadherin cleavage.

We show the protein cleavage data in **Supplementary Figure S11** and discuss it on **lines 380-389**.

G. What is DC assay?

Response 12: DC assay is **Detergent Compatible** protein assay. It is a standard way to measure protein concentration of cell lysate. We have added the full name of this assay to the text before first using this abbreviation (**line 269**).

H. Finally, it would be beneficial if authors provide additional experimental supports for the model of stretched vs relaxed cell-cell junction;

(1) Strength of cell-cell junction with/without CQY while arresting/blocking internalization.

Response 13: As described in Response 9, we have provided additional data showing that X-dimer formation (either mediated by CQY684 binding or via the W2A mutation) triggers p120 phosphorylation and dissociation from the Pcad cytoplasmic region, which promotes Pcad internalization. Although conformational changes in the Pcad cytoplasmic region due to stretching of the cell junction may be one mechanism for phosphorylation of p120, many alternate mechanisms exist include kinases that specifically recognize X-dimer and phosphorylate p120-catenin.

We now discuss these alternate mechanisms on **lines 431-438**. We have also de-emphasized the discussion of stretched vs. relaxed cell-cell junctions and moved the data to the SI (**Supplementary Figure S13**). We respectfully believe that identifying the exact mechanism by which p120 gets phosphorylated is beyond the scope of this work.

While it would be interesting to assess cell-cell junction strength with/without CQY684 while blocking Pcad internalization, there is currently no established method to selectively inhibit Pcad internalization. Previous studies have identified mutations in the cytoplasmic region of Cad11 that block its internalization¹², but these mutations may not apply to Pcad. Furthermore, the use of chemical inhibitors that block all protein internalization would complicate the interpretation of our results, as it would be difficult to distinguish the effects of CQY684 mediated strengthening of X-dimers from effects that inhibit other internalized proteins. For these reasons, we respectfully submit that the experiments suggested by the reviewer are beyond the scope of this study.

There are certain minor comments related to referencing and the comments are directly made on the manuscript.

Response 14: We thank the reviewer for suggesting these references and have incorporated them into the manuscript.

Reviewer #2 (Remarks to the Author):

This study by Xie et al combines single molecule experiments, simulations, and cell-based experiments to determine the mechanism of action of the PCadherin binding antibody CQY. This multi-faceted multi-scaled approach is a strong way to address this important complex biophysical and cell biological question. Understanding this mechanism is important for both drug delivery

and basic cell biology, making it of great interest. However the results are not supported as key controls and orthogonal validations are lacking and details of some experiments and statistical tests are not well described. Statistical methods are not well described. At times the results jump, referencing experiments that have not yet been described which is difficult to follow. These were the main

A key conclusion of this study is that CQY binding traps Pcad in an X-dimer, leading to dissociation of p120 which promotes PCad endocytosis and degradation.

We thank the reviewer for their suggestions. We have now performed additional experiments and analysis to address all the reviewer's comments

The data suggesting less PCad on the surface is Fig4d. This experiment lacks controls, at a minimum positive and negative controls for the assay as well as a surface protein not impacted by CQY should also be probed.

Response 15: We thank the reviewer for this suggestion. We have now included a negative control (tubulin, a cytoplasmic protein) and a positive control (CD98, a membrane protein) in our surface biotinylation assay (Figure R6). As can be seen in the figure, tubulin was not detected in the surface fraction whereas CD98 was detected to a similar extent in both the surface fraction and the lysate. This data has been included in **Figure 4d** of the manuscript.

Figure R6. Surface biotinylation experiments were validated using positive and negative controls. Tubulin and CD98 expression in lysate and biotinylated surface fractions were detected using corresponding antibodies. As expected, tubulin (serving as the negative control) was not detected in the surface fraction, whereas CD98 was detected to a similar extent in both the surface fraction and the lysate.

The elute/lysate is confusing - in the text 40% is mentioned but this plot shows 100%.

Response 16: We apologize for this confusion. In our original plots, we had normalized all the elute/lysate data against the -CQY conditions. Consequently, all the data in the bar chart was ~100%. The reviewer's comment made us realize that this was confusing and so we have now displayed the data without normalization. Consequently, the text and plots (**Figure 4d**) are now consistent in terms of the fraction of total Pcad expressed on the cell surface. Additionally, to avoid a similar confusion in other figures, we have added "Norm." in the y-axis label when we plot normalized data.

That there is not a change in ratio between total Pcad and surface in the 2 treatment groups is surprising seeing the microscopy. Along with the controls, an orthogonal experiment demonstrating there is less PCad on the cell membrane would strengthen the conclusions.

Response 17: Respectfully, we do not believe there is an inconsistency between our surface biotinylation and imaging results. Our surface biotinylation experiments clearly demonstrate that the surface level of Pcad in WT cells decrease upon CQY684 treatment (Figure 4d) which is consistent with the imaging data (Figure 4c). We would like to highlight that since surface biotinylation assays are amongst the most widely used methods for quantifying surface protein expression levels, we used this approach in our experiments.

We believe the reviewer's concern arises from the data showing that the ratio between total Pcad and surface Pcad is the same in all conditions. However, this is because the total amount of Pcad in the cell is reduced upon CQY binding, due to increased lysosomal proteolysis. Imaging data and western blot results (**Supplementary Figure S11**) show an increased internalization and lysosomal proteolysis of Pcad upon CQY treatment. We show the protein cleavage data in **Supplementary Figure S11** and discuss it on **lines 380-389**.

The data suggesting there is less p120 binding to pcad is presented in Fig4e. It is not clear if the reduction in p120 binding reflects the smaller proportion of pcad present on the cell membrane or less p120 binding for the pcad that is at the membrane.

Response 18: The reviewer is correct in pointing out that the reduction of p120 signals could result from decreased Pcad levels on the cell membrane, leading to less p120 being pulled down in Co-IP experiments. To address this, **we normalized the p120 levels to the amount of pulled-down Pcad (p120/Pcad), as shown in Figure 5b (previously Figure 4e).**

The control box is completely empty, this cannot be interpreted.

Response 19: We used rabbit IgG as a negative control since it shouldn't pull down any protein from the lysate. Consequently, the control box is completely empty.

The authors refer to the elute/lysate in Fig4e as justification for the conclusion. However this data needs controls (see above) and seems to counter the imaging which shows significantly more internalized pcad in the +CQY cells. This should be validated with orthogonal experiments such as p120 staining. Can biochemistry be used to pull down only CQY bound pcad and probe p120 binding?

Response 20: As suggested by the reviewer, we performed a p120 immunostaining experiment. Briefly, we immunostained for Pcad and p120 and used ImageJ to identify Pcad signal spots that also displayed p120 expression (Figure R7). We normalized the fraction of p120-associated Pcad by the fraction of full-length Pcad in -CQY condition and +CQY condition (obtained from Supplementary Figure S11). This is an important correction since we only measured the fraction of p120 associated with full length Pcad in our Co-IP experiments. While cell lysate western blots

(Supplementary Figure S11) show both a band corresponding to the full length Pcad and bands corresponding to proteolytic fragments of Pcad, we are unable to distinguish Pcad fragments from full length Pcad in the immunofluorescence images. Our immunofluorescence analysis cross-validate our Co-IP results by showing that the fraction of p120 associated to full-length Pcad was higher in the -CQY condition (39.3%) compared to the +CQY condition (33.5%) (Figure R7).

Despite this qualitative agreement, we believe that the immunofluorescence results presented in Figure R7 need to be interpreted very cautiously due to the following reasons:

- (i) From the immunofluorescence images it is impossible to distinguish non-functional fragments of Pcad from functional full length Pcad. For instance, if we assume that all the Pcad in the immunofluorescence image are full-length, the Pearson's colocalization coefficients between Pcad and p120 are not significantly different between the -CQY and +CQY conditions.
- (ii) p120 can bind to other cadherins in A431 cells such as N-cadherin^{13,14} and desmosomal cadherins¹⁵. Furthermore, similar to β -catenin which can bind directly to the membrane¹⁶, p120 may also directly associate with the membrane compartment. As can be seen in Figure R7, a significant fraction of p120 remained localized at the cell membrane in the +CQY condition.

Given these caveats, we respectfully believe that immunostaining is not a robust orthogonal experiment to cross-validate functional full-length Pcad/p120 association. We have therefore chosen not to include this image in the SI.

Figure R7. Immunostaining cross-validates Co-IP experiments. Left panel: Representative images of Pcad and p120 staining with or without CQY treatment. Right panel: The fraction of full-length p120 associated Pcad with respect to total Pcad in -CQY and +CQY conditions. In the -CQY condition, 39.3% \pm 5.1% of Pcad was associated with p120, while in the +CQY condition, 33.5% \pm 5.0% was associated.

The authors should show that at the timepoints of their experiments CQY is still bound to the cell surface.

Response 21: To address the reviewer's concern, we stained CQY at the time point of our experiments (Figure R5). Despite significant internalization of CQY, a fraction of the antibody remained bound to the cell surface, likely because of the excess free CQY added to the buffer.

We now present this data in **Supplementary Figure S8** and discuss the data on **lines 238-240** of the manuscript.

Figure R8. CQY684 remains bound at the cell surface at experimental timepoints. Staining of Pcad (left) and CQY684 (right) after overnight incubation with CQY684. A fraction of CQY684 remains bound on the cell surface.

Critically the concentration of CQY is not reported for multiple experiments in the legend or Materials and Methods. When p values are reported, the statistical test used is not included and often it is challenging to determine what was considered “n”. Some p values seem extremely small.

Response 22: We thank the reviewer for their feedback. We have now added units after the N in the figure legend, for example, N =150 cells. The statistical test we used is two-sided student T-test throughout the paper and we now explicitly mention this in the Materials & Method section. The concentration of CQY for different experiments are now reported in the legend and Materials & Methods section. The p values are small due to the significant statistical differences between our test sample groups.

Please mention any implications in the modeling for using the PCad Xdimer +CQY but simply removing the CQY Fabs (P7L18-19)?

Response 23: There are two reasons we used this strategy. First, by removing only the Fab, we can directly compare the effects of CQY684 on the same initial X-dimer structure, avoiding potential bias from selecting a different starting structure. Second, besides this structure (PDB: 6ZTR), we were unable to find a complete Pcad X-dimer structure in the PDB, as all available Pcad X-dimer structures have missing residues on the Pcad EC1/EC2 domains.

Could the antibody binding change the X-dimer conformation?

Response 24: As described in Responses 1 and 3, antibody binding slightly changes the X-dimer conformation after the system equilibrates in the MD simulation. Briefly, the slight change in the X-dimer conformation introduces a previously non-persistent salt bridge, which contributing

significantly to the strengthening of X-dimer interactions. The changes in the interactions are explained in detail in responses 1 and 3.

Described below are some additional questions.

Figure 1 What is considered the “n” in these experiment?

Response 25: N represents the number of single molecule unbinding events measured during the AFM experiments. For bead aggregation assays, N represents the number of bead aggregates observed. We now explicitly state the ‘n’ in the figure legend.

Figure 4b what is “n”? And what is the y axis (cell counts) in the context of this experiment, I understood it to be an aggregation experiment??

Response 26: The "n" refers to the number of cell aggregates observed, and we now mention this in the figure legend. For each aggregate, we counted the number of cells within it, which is referred to as the cell count. The idea is that larger aggregates, with higher cell counts, indicate stronger cell adhesion, as more cells are involved in forming a single aggregate.

Fig 4c Co-localization. Is this a single z plane? The LAMP1 staining does not look the same between the +/- CQY conditions. Indeed, there is hardly any LAMP-1 in the control condition.

Response 27: Yes, this image corresponds to a single z-plane. To directly address the reviewer’s comment, we measured the integrated LAMP1 intensity within individual cells (Figure R6). We found that the mean intensity of LAMP1 signals within the cytoplasmic region is the same for both the +CQY and –CQY conditions. While there are similar amounts of LAMP1 under both conditions, in the absence of CQY684 treatment, the LAMP1 signal is more dispersed compared to cells treated with CQY684.

We now include this data in **Supplementary Figure S9** and describe it on **lines 266-267**.

Figure R9. Mean LAMP1 signal intensity was similar in cells with and without CQY treatment. For the -CQY condition, the average LAMP1 intensity was 13.80 ± 4.45 , and for the +CQY condition, the average LAMP1 intensity was 13.67 ± 3.78 . The P-value of 0.84 indicates no significant difference between these two conditions.

Are the images acquired and scaled in the same way so intensities are comparable? Additionally, the images appear to be scaled differently looking at the 10 μm scale bars.

Response 28: For the co-localization analysis, we specifically acquired images in a single z-plane. During the imaging session, the laser intensity, pinhole size, and smart gains were set to the same values, and the pixel size for each image was adjusted to approximately 20 μm x 20 μm to ensure comparability of the results. To present the effects of CQY, we zoomed in on the cell cluster regions, which resulted in varying lengths of the 10 μm scale bars. We have added additional methodological description in the method section, **lines 578-581**.

What do the “n” in the figure legend correspond to? Number of images acquired?

Response 29: The 'n' number in the figures refers to the number of cells analyzed for the Pearson's coefficient plot, not the number of images. We now mention this in the figure legend

Reviewer #3 (Remarks to the Author)

This work used a combination of biophysical, computational, biochemical and cell biological methods to demonstrate that an antibody, CQY684, can trap P-cadherin in its X-dimer conformation, which can trigger P120 dissociation and lead to the endocytosis of P-cadherin. These discoveries offer a previously unknown mechanism of outside-in signaling and provide new insights to the drug delivery. The overall research is well designed, and the results are highly interesting. I recommend the acceptance of the manuscript to Nature Communications after the authors provide the following justifications.

We thank reviewer #3 for their positive feedback. We have addressed the reviewer's comments below

1. the construction of the complex formed by P-cadherin X-dimer and antibody needs more details.

Response 30: For our simulations, the complex formed by the Pcad X-dimer and antibody was generated using the PDB structure (PDB code 6ZTR). We include details on how the model was generated on **lines 504-507** of the methods section.

The original crystal structure only contains cadherin's first two N-terminal domains and the antibody. Is it possible that the antibody could also bind to the cadherin in its S-dimer configuration?

Response 31: S-dimer formation is mediated by exchange of a conserved Trp (W2) on the EC1 domain which is part of the structure in our simulation. We also aligned the bound CQY to the Pcad S-dimer and found that it does not inhibit S-dimer formation (Figure R10). Since this finding has already been reported in a previous study¹⁷, we decided not include the alignment as a supplementary figure. We however describe this on **lines 369-373** of the manuscript.

Figure R10. CQY684 binding does not interfere with Pcad S-dimer interface. CQY684 was aligned on a Pcad S-dimer (colored in green and cyan, PDB code: 4ZML). The binding site for the antibody and the S-dimer are well separated and no steric clashes are observed.

2. This study doesn't mention another important interaction, the cis-interaction, which is critical to the formation of junction. The antibody binding interface overlaps with the cis binding interface, based on the observation. Could the author add justification to discuss how cis interaction could be affected with the presence of antibodies?

Response 32: We thank the reviewer for highlighting the role of *cis*-interactions in junction formation. We aligned CQY684 on the Pcad *cis*-dimer structure and found that CQY684 binding does not affect Pcad *cis*-interactions. These results are shown in **Supplementary Figure S10**. We have included a discussion about this on **lines 375-378**.

3. In addition to the outside-in signaling, the inside-out signaling has also recently been proposed to the cadherin-p120 system, in which the binding of p120 to the cytoplasmic region of cadherin could change the configuration of its extracellular domains, changing its intercellular trans or cis interactions, and thus regulating cell adhesion. The authors should mention this in either introduction or discussion.

Response 33: We thank the reviewer for bring up this important point. We now discuss p120 inside-out signaling and cite corresponding references on **lines 64-66** of the introduction section, and on **lines 422 to 430** of the discussion section.

4. The SMD simulation of Fig 5b does not provide direct evidence to how x-dimer formation leads to dephosphorylation of p120, and it's kind of straightforward that proteins would dissociate if being pulled apart. Therefore, the authors could either remove this simulation part or put it in the supplementary material.

Response 34: We thank the reviewer for this excellent suggestion. We have eliminated mention of stretched vs. relaxed cell-cell junctions from the manuscript and moved the data to the SI (**Supplementary Figure S13**).

References

- 1 Kudo, S., Caaveiro, J. M. & Tsumoto, K. Adhesive Dimerization of Human P-Cadherin Catalyzed by a Chaperone-like Mechanism. *Structure* **24**, 1523-1536, doi:10.1016/j.str.2016.07.002 (2016).
- 2 Kumar, S. & Nussinov, R. Relationship between ion pair geometries and electrostatic strengths in proteins. *Biophys J* **83**, 1595-1612, doi:10.1016/S0006-3495(02)73929-5 (2002).
- 3 Harrison, O. J. *et al.* Two-step adhesive binding by classical cadherins. *Nat. Struct. Mol. Biol.* **17**, 348-357, doi:10.1038/nsmb.1784 (2010).
- 4 Maker, A. *et al.* Regulation of multiple dimeric states of E-cadherin by adhesion activating antibodies revealed through Cryo-EM and X-ray crystallography. *PNAS Nexus* **1**, pgac163 (2022).
- 5 Nishiguchi, S., Furuta, T. & Uchihashi, T. Multiple dimeric structures and strand-swap dimerization of E-cadherin in solution visualized by high-speed atomic force microscopy. *Proc Natl Acad Sci U S A* **119**, e2208067119, doi:10.1073/pnas.2208067119 (2022).
- 6 Li, Y. *et al.* Mechanism of E-cadherin dimerization probed by NMR relaxation dispersion. *Proc Natl Acad Sci U S A* **110**, 16462-16467, doi:10.1073/pnas.1314303110 (2013).
- 7 Hong, S. J., Troyanovsky, R. B. & Troyanovsky, S. M. Cadherin exits the junction by switching its adhesive bond. *J. Cell Biol.* **192**, 1073-1083, doi:10.1083/jcb.201006113 (2011).
- 8 Kourtidis, A., Yanagisawa, M., Huvelde, D., Copland, J. A. & Anastasiadis, P. Z. Pro-Tumorigenic Phosphorylation of p120 Catenin in Renal and Breast Cancer. *PLoS One* **10**, e0129964, doi:10.1371/journal.pone.0129964 (2015).
- 9 Anastasiadis, P. Z. & Reynolds, A. B. The p120 catenin family: complex roles in adhesion, signaling and cancer. *J Cell Sci* **113 (Pt 8)**, 1319-1334, doi:10.1242/jcs.113.8.1319 (2000).
- 10 Miyashita, Y. & Ozawa, M. Increased internalization of p120-uncoupled E-cadherin and a requirement for a dileucine motif in the cytoplasmic domain for endocytosis of the protein. *J Biol Chem* **282**, 11540-11548, doi:10.1074/jbc.M608351200 (2007).
- 11 Ishiyama, N. *et al.* Dynamic and Static Interactions between p120 Catenin and E-Cadherin Regulate the Stability of Cell-Cell Adhesion. *Cell* **141**, 117-128, doi:10.1016/j.cell.2010.01.017 (2010).
- 12 Satcher, R. L. *et al.* Cadherin-11 endocytosis through binding to clathrin promotes cadherin-11-mediated migration in prostate cancer cells. *J Cell Sci* **128**, 4629-4641, doi:10.1242/jcs.176081 (2015).

- 13 Daniel, J. M. & Reynolds, A. B. The catenin p120(ctn) interacts with Kaiso, a novel BTB/POZ domain zinc finger transcription factor. *Mol Cell Biol* **19**, 3614-3623, doi:10.1128/MCB.19.5.3614 (1999).
- 14 Taulet, N. *et al.* N-cadherin/p120 catenin association at cell-cell contacts occurs in cholesterol-rich membrane domains and is required for RhoA activation and myogenesis. *J Biol Chem* **284**, 23137-23145, doi:10.1074/jbc.M109.017665 (2009).
- 15 Carnahan, R. H., Rokas, A., Gaucher, E. A. & Reynolds, A. B. The molecular evolution of the p120-catenin subfamily and its functional associations. *PLoS One* **5**, e15747, doi:10.1371/journal.pone.0015747 (2010).
- 16 Xu, W. & Kimelman, D. Mechanistic insights from structural studies of beta-catenin and its binding partners. *J Cell Sci* **120**, 3337-3344, doi:10.1242/jcs.013771 (2007).
- 17 Sheng, Q. *et al.* PCA062, a P-cadherin Targeting Antibody–Drug Conjugate, Displays Potent Antitumor Activity Against P-cadherin–expressing Malignancies. *Molecular Cancer Therapeutics* **20**, 1270-1282, doi:10.1158/1535-7163.Mct-20-0708 (2021).